# Deployment expectations of multi-gigatonne scale carbon removal could have adverse impacts on Asia's energy-water-land nexus

Jeffrey Dankwa Ampah [1,2], Chao Jin[1] ✉, Haifeng Liu [2] ✉, Mingfa Yao[2] ✉, Sandylove Afrane[1], Humphrey Adun [3], Jay Fuhrman [4], David T. Ho [5,6] & Haewon McJeon[7]

Existing studies indicate that future global carbon dioxide ($CO_2$) removal (CDR) efforts could largely be concentrated in Asia. However, there is limited understanding of how individual Asian countries and regions will respond to varying and uncertain scales of future CDR concerning their energy-land-water system. We address this gap by modeling various levels of CDR-reliant pathways under climate change ambitions in Asia. We find that high CDR reliance leads to residual fossil fuel and industry emissions of about 8 Gigatonnes $CO_2yr^{-1}$ ($GtCO_2yr^{-1}$) by 2050, compared to less than 1 $GtCO_2yr^{-1}$ under moderate-to-low CDR reliance. Moreover, expectations of multi-gigatonne CDR could delay the achievement of domestic net zero $CO_2$ emissions for several Asian countries and regions, and lead to higher land allocation and fertilizer demand for bioenergy crop cultivation. Here, we show that Asian countries and regions should prioritize emission reduction strategies while capitalizing on the advantages of carbon removal when it is most viable.

The window for achieving the Paris Agreement's aspirational goal of limiting global temperature change to 1.5 °C above pre-industrial levels is rapidly closing. At the current rates, the 1.5 °C remaining carbon budget could be completely exhausted in the late 2020 s[1]. Immediate and effective decarbonization measures, such as replacing fossil fuels with renewables, electrification, energy efficiency improvement, and carbon-neutral fuels, are required on a large scale to reduce greenhouse gas (GHG) emissions and ultimately address climate change. While these measures are crucial for reducing emissions, they are essentially preventive and insufficient[2–4] to achieve net-zero carbon dioxide ($CO_2$) or GHG emissions consistent with limiting warming to below 1.5 °C by 2100.

Carbon dioxide removal (CDR) technologies have gained significant attention in recent years and effectively contribute to putting the "net" in "net-zero emissions"[5]. CDR does not have to replace significant emission reductions but could play a key role in counterbalancing residual emissions from hard-to-abate sectors at the point of net zero[2]. Despite the importance of CDR in aiding the realization of global net zero targets, they could present adverse impacts on global energy-land-water system[6], especially when there is over-reliance on them for "solving" climate change[7]. For instance, bioenergy with carbon capture and storage (BECCS) is land- and water-intensive[8,9], while direct air capture with carbon storage (DACCS) is highly energy-intensive and can increase emissions[10,11]. Previous studies[10,12–14] show that more than 10 Gigatonnes $CO_2yr^{-1}$ ($GtCO_2yr^{-1}$) of gross CDR would have to be deployed by mid-century to limit warming to below 1.5 °C by 2100. Considering the potential trade-offs and spillover effects associated with different CDR methods, an open question remains - that is, quantitatively speaking, what could be the extent of damage to a

[1]School of Environmental Science and Engineering, Tianjin University, Tianjin, China. [2]State Key Laboratory of Engines, Tianjin University, Tianjin, China. [3]Operational Research Centre in Healthcare, Near East University, Nicosia, Turkey. [4]Joint Global Change Research Institute, University of Maryland and Pacific Northwest National Laboratory, College Park, MD, USA. [5]Department of Oceanography, University of Hawaii at Mānoa, Honolulu, HI, USA. [6][C]Worthy, Boulder, Colorado, USA. [7]KAIST Graduate School of Green Growth & Sustainability, Daejeon, Korea. ✉e-mail: jinchao@tju.edu.cn; haifengliu@tju.edu.cn; y_mingfa@tju.edu.cn

region or country's energy-land-water system an expected multigigatonne CDR deployment by mid-century?

Using a modified version of the Global Change Assessment Model (GCAM-TJU), we explore four main scenarios in this study. The first scenario represents a high reliance on CDR, which we label "HIGH." In this scenario, six different CDR approaches are deployed in Asia as early as 2025, with no limit on the amount of $CO_2$ that can be removed from the atmosphere at any point in time. The second scenario represents a moderate reliance on CDR, which we label "MODERATE." In this scenario, only BECCS (novel CDR) is deployed alongside afforestation/reforestation (AR; conventional CDR) in Asia from 2025, and we restrict $CO_2$ removal by BECCS to an annual average of 1.8 $GtCO_2yr^{-1}$. The third scenario is the "LOW" scenario, representing a future where Asia does not rely on any novel means of removing $CO_2$ from the atmosphere. In this scenario, the source of negative emissions is solely through AR. The final scenario is the "REFERENCE" scenario, where Asia does not develop any new climate policies over the century. Asia's fossil fuel consumption and emissions are expected to continue growing under this scenario (see Table 1 and "Methods" section).

In this work, we make the following contributions to the existing literature. First, studies that currently quantify the energy-land-water trade-offs associated with varying levels of CDR reliance remain scarce. To date, the literature on the adverse impacts of high reliance on CDR has been primarily philosophical, theoretical, and qualitative[15–20]. Also, policy formulation based on existing pathways often treats conventional emission reduction and CDR as equivalent or interchangeable mitigation options[21,22] under an optimal carbon price. This approach continues to defer urgent emission cuts under the assumption of future large-scale removals[23]. There are thus urgent calls to separate CDR and emission reduction targets[24,25], and responding to these calls, we model pathways where CDR only scales to the amount specified and does so without undermining emission cuts. Moreover, in previous cases where CDR has been limited, modelers often resorted to decreasing carbon capture and storage (CCS)[26,27] and biomass supply[12,28] as a way to slow down CDR deployment. Here, we have explicitly set caps on actual CDR at each modeling period without affecting CCS and biomass supply needed for other activities beyond negative emissions. In addition, we have focused on the Asian region, where a significant share of future global CDR could be concentrated without equity considerations[12,29,30]. While few quantitative studies exist on the impact of high CDR (mainly on the global energy system)[30–33], quantitative analysis on other sustainability factors such as land and water implications under varying/uncertain future carbon removal remains limited. Furthermore, existing studies continue to predominantly model land-based CDR approaches (such as BECCS and AR) globally[34–40], and countries with land and biomass limitations are at risk of completely realizing their CDR targets with such approaches[29].

Here, we demonstrate how six different CDR types (land-based, chemical, and geochemical) are distributed across Asia, and which ones could be more suitable for countries or regions to pursue towards achieving their respective climate ambitions.

With Asia poised to become the region with the highest future carbon removal capacity (without equity considerations), we model different stylized CDR pathways for the region and explore how they would impact the region's emission reduction goals. Our analysis focuses on the energy, water, land, mitigation costs, and air pollution impacts of these CDR pathways, providing critical insights for policymakers to balance emission reductions and realistic negative emissions targets.

## Results

### Impact on primary and final energy demand, and abatement costs

The remaining carbon budget can be extended by attempting to remove large quantities of $CO_2$ from the atmosphere[41], delaying the necessary transition away from fossil fuels to achieve net zero targets. Our results indicate that high reliance on CDR presents a risk of perpetuating fossil fuel consumption and impedes transitions to cleaner alternatives (Fig. 1). In contrast, lower reliance on negative emissions leads to higher supply of cleaner energy sources (e.g., renewables and nuclear energy) and increased electrification and energy-efficiency improvement in end-use sectors such as buildings and transport. Our results indicate that by 2050, the share of unabated fossil fuel consumption in total primary energy supply in Asia could reach over 30% under HIGH CDR scenario compared to less than 10% under MODERATE and LOW CDR scenarios. Moreover, at least an additional 70 Exajoules per year ($EJ\ yr^{-1}$) of coal and natural gas would be consumed in Asia by 2050 in end-use sectors under HIGH CDR scenario compared to consumptions under MODERATE and LOW CDR scenarios (more details in Supplementary Table 1).

Detailed discussions for primary energy consumption, electricity and hydrogen production by source from 2025-2050 for different levels of CDR reliance in Asia alongside a no-climate-policy scenario are summarized in Supplementary Discussion 1.

Final energy consumption is only 5% lower in the HIGH CDR scenario than in the REFERENCE scenario in 2050. Under relatively low reliance on CDR, fossil fuel consumption, inefficient practices, and technologies are rapidly phased down. As a result, the MODERATE and LOW CDR scenarios show reductions of 35% in final energy consumption compared to the REFERENCE scenario (Supplementary Fig. 1). The marginal abatement cost of carbon, also known as the $CO_2$/GHG price, refers to the cost incurred in reducing the last unit (e.g., one tonne) of $CO_2$/GHG emissions[2]. It represents the additional cost required to achieve an incremental reduction in emissions. Under limited availability of carbon removal, the marginal abatement cost of

## Table 1 | Scenario formulation and description

| Scenario | Description | CDR availability | Constraint on CDR |
|---|---|---|---|
| Central | A stylized pathway where Asia reaches net zero GHG emissions by mid-century | – | – |
| HIGH | Follows the central pathway with the availability of six different CDR approaches | AR, Biochar, BECCS, DACCS, DORCS, and ERW | No constraint. Total gross $CO_2$ removal reaches 12 $GtCO_2yr^{-1}$ by 2050 |
| MODERATE | Follows central pathway with two CDR approaches | BECCS and AR | Constraint on BECCS deployment to an annual average of 1.8 $GtCO_2yr^{-1a}$ from 2025 to 2050. Total gross $CO_2$ removal reaches 2.3 $GtCO_2yr^{-1}$ by 2050 |
| LOW | Follows central pathway with the availability of one CDR approach | AR only | Constraint on novel CDR at 0 $GtCO_2yr^{-1}$ from 2025 to 2050. Total gross $CO_2$ removal reaches 0.5 $GtCO_2yr^{-1}$ by 2050 |
| REFERENCE | No new climate mitigation policy | AR | No |

*AR* afforestation and reforestation, *BECCS* bioenergy with carbon capture and storage, *DACCS* direct air capture and carbon storage, *DORCS* direct ocean removal and carbon storage, *ERW* enhanced rock weathering.
aTotal current amount of global CDR deployment is 2 $GtCO_2yr^{-1}$.

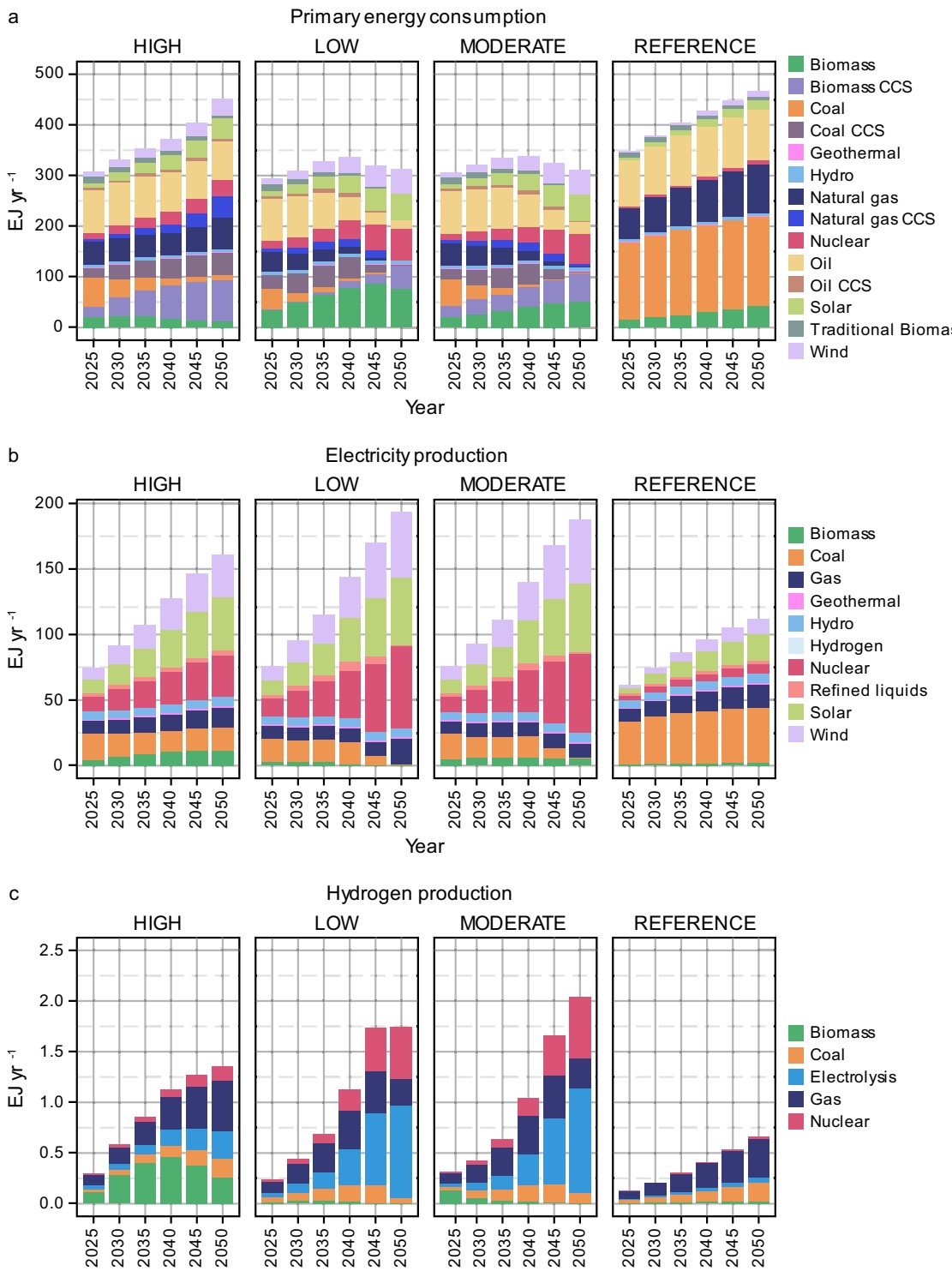

**Fig. 1 | Impacts on Asia's energy system.** Primary energy consumption by source (**a**). Electricity production by source (**b**). Hydrogen production by source (**c**). All results are from 2025 to 2050 for different levels of carbon dioxide removal (CDR) reliance (HIGH, MODERATE, and LOW) in Asia alongside a no-climate-policy scenario (REFERENCE). CCS carbon capture with storage, EJyr⁻¹ exajoule per year.

carbon is considerably higher in the MODERATE and LOW CDR scenarios than in the HIGH scenario (Supplementary Fig. 2a and Supplementary Table 1). CDR technologies, while expensive, have the potential to delay the urgent need for emission reduction[42]. In particular, DACCS can serve as a backstop technology for climate change mitigation, providing a last-resort option to reduce $CO_2$ emissions

when other mitigation options especially in hard-to-abate sectors become prohibitively expensive or impractical[13]. Without CDR, the marginal abatement cost curve tends to increase exponentially as we approach 100% mitigation[43], indicating that costs escalate rapidly as emissions reduction targets become more stringent. This arises from greater reliance on expensive technologies to eliminate the remaining

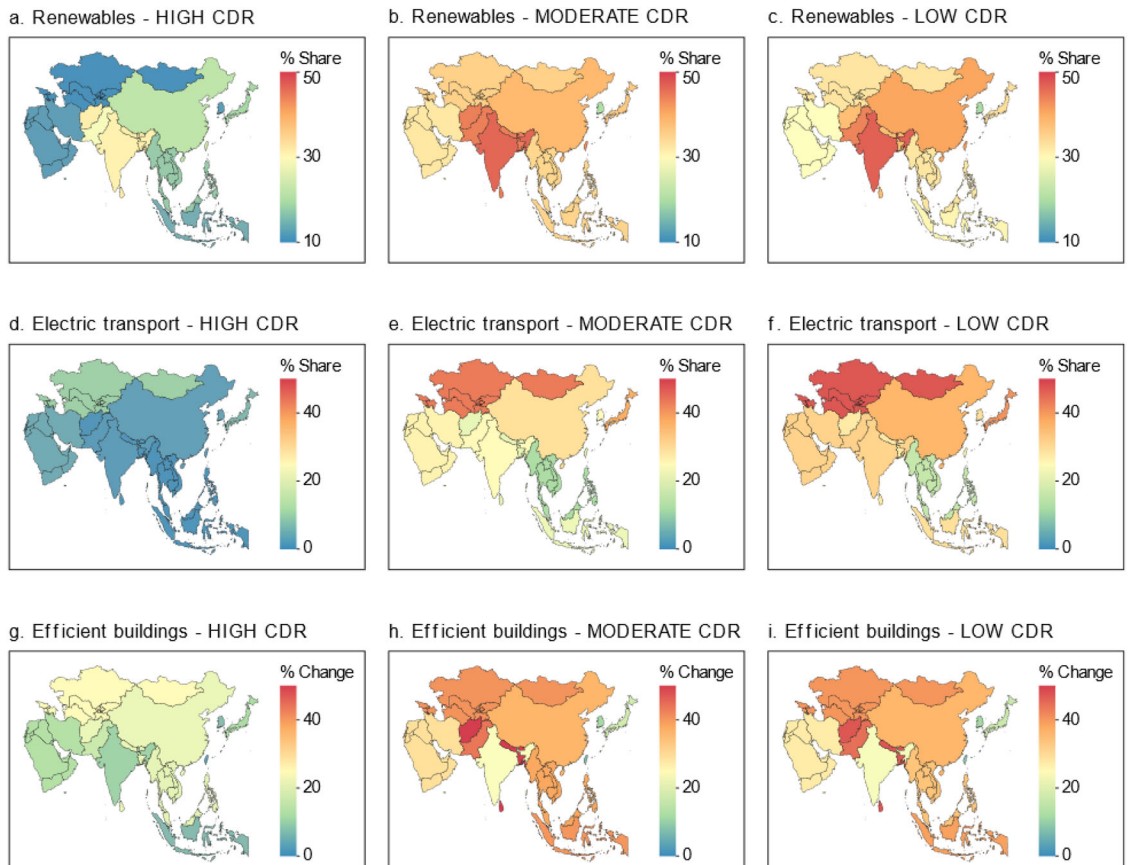

**Fig. 2 | Response of Asian countries and regions to energy sector transition under varying carbon dioxide removal (CDR) reliance.** The results include the share of renewables in total primary energy consumption by 2050 (**a**–**c**), the share of electrified transport in the transport sector by 2050 (**d**–**f**), and building final energy consumption reduction relative to a reference scenario by 2050 (**g**–**i**). The findings indicate that under the HIGH CDR scenario, the highest share of electrified transport in any Asian region or country remains below 15%, with Central Asia and Japan having the highest shares at 12.90% and 8.64%, respectively. Countries such as China and India exhibit shares of only 4.50% and 3.75% for electrified transport,

respectively. Under the MODERATE and LOW CDR scenarios, the highest share of electrified transport can rapidly increase to 42–47%, with China and India's shares reaching about 25–30% and 30–35%, respectively. Under the HIGH CDR scenario (in contrast to the REFERENCE case), Central Asia, South Asia, and China would record the highest reductions in building energy consumption by 2050 at 25.39%, 22.6%, and 22.50%, respectively. These reduction rates significantly increase to 38.8%, 50%, and 34% in these regions, respectively, under the MODERATE CDR scenario. Our analysis considered Island countries/regions and Territories in Asia but they are not included in the map.

emissions. Without CDR in Asia's net zero strategies, capital-intensive cleaner technologies and fuels with increasingly low utilization rates compared to their fossil fuel counterparts would need to expand substantially to enable significant emission cuts[44]. This ultimately increases the marginal costs, making the LOW and MODERATE CDR scenarios the most expensive pathways explored here. Low carbon price under CDR-based pathways is also widely reported in several studies[13,30,43,44].

The results are further emphasized spatially in the Asian regions and countries (Fig. 2). The findings indicate that countries, such as China and India, could satisfy their primary energy demands in 2050, with 33% and 45% from renewables (wind, hydro, geothermal, and solar) under the MODERATE CDR scenario, compared to 17% and 27% under the HIGH CDR scenario, respectively. In the HIGH CDR scenario, share of electrified transport remains relatively low across Asia, with none of the countries or regions exceeding 15%. Countries such as China and India have particularly lower shares of electrified transport, which could be attributed to their carbon-intensive transport systems. In contrast, the share of electrified transport will witness significant growth under the MODERATE and LOW CDR scenarios. China and India's shares can increase to approximately 25-35%. Similarly, there is a marked reduction in building energy consumption by 2050 in the

LOW and MODERATE CDR scenarios compared to the HIGH CDR scenario.

## Impact on positive and negative emissions, net zero timing, and air pollutants

Economically, undue substitution is a typical characteristic of high CDR pathways, allowing for continued fossil emissions through carbon offsetting[45,46]. Total positive GHG emissions (including fossil and industry, and biogenic $CO_2$) in 2050 are lower in the MODERATE (11 $GtCO_2eyr^{-1}$) and LOW (13 $GtCO_2eyr^{-1}$) CDR scenarios than in the HIGH CDR scenario (16 $GtCO_2eyr^{-1}$) (Supplementary Fig. 2b). Excluding biogenic $CO_2$, total GHG emissions by 2050 reaches 12.95, 5.38, and 5 $GtCO_2eyr^{-1}$ under HIGH, MODERATE, and LOW CDR scenarios, respectively. The slow phase down of fossil fuels under HIGH CDR scenario results in slower decarbonization across all energy sectors. By 2050, over 2 $GtCO_2yr^{-1}$ and 3 $GtCO_2yr^{-1}$ of fossil fuel and industry (FFI) emissions will remain in Asia's industrial and transport sectors, respectively under multi-gigatonne CDR pathway (HIGH CDR scenario). Rapid uptake of electrification, carbon-neutral fuels, and energy efficiency improvement under lower CDR pathways (MODERATE and LOW CDR scenarios) would ensure that residual FFI emissions from each of these two sectors stay below 0.5 $GtCO_2yr^{-1}$ by 2050 (more

a

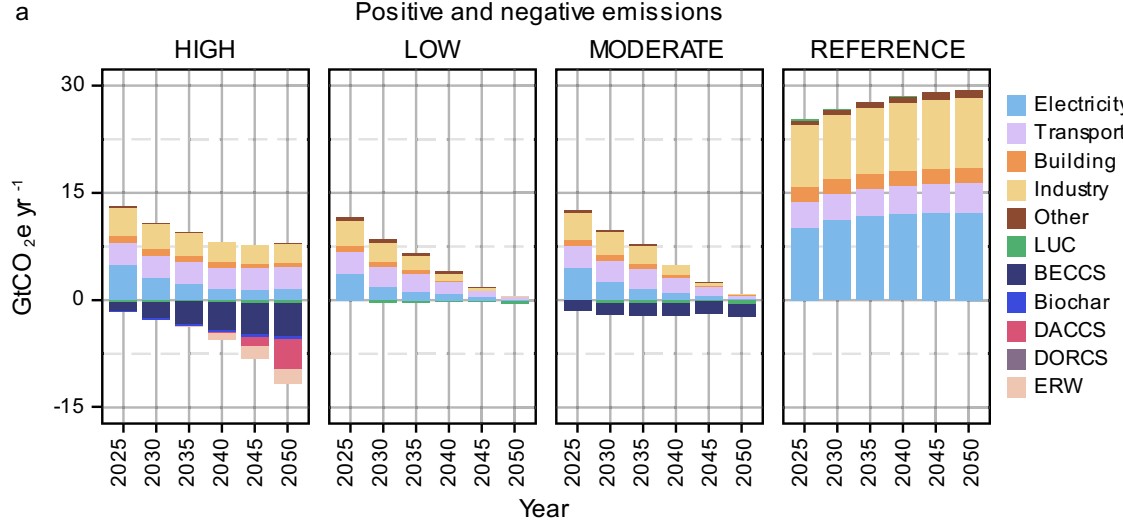

b

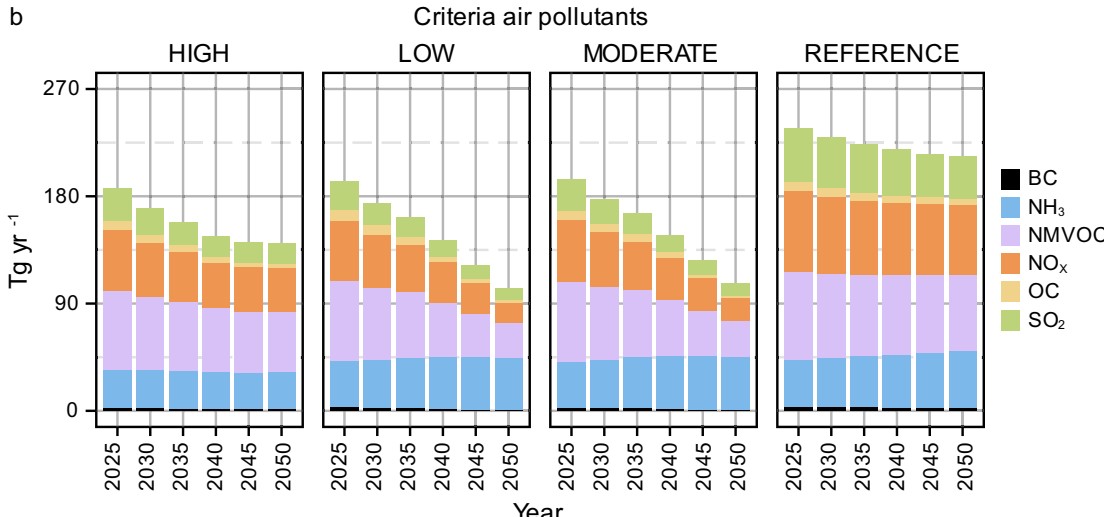

**Fig. 3 | Impacts on emissions and air pollution.** Total positive and negative $CO_2$ emissions by sector and species (**a**). It should be noted that Asia reaches net zero GHG emissions around mid-century, indicating net zero $CO_2$ and net negative $CO_2$ emissions could occur in earlier years depending on the scenario and the amount of carbon dioxide removal (CDR) available to offset residual emissions. Positive values for LUC in some years represent net positive emissions, where deforestation exceeds the afforestation rate. Total air pollutants in Asia (**b**). All results are from 2025-2050 for different levels of CDR reliance (HIGH, MODERATE, and LOW) in Asia alongside a no-climate-policy scenario (REFERENCE). "Other" refers to other energy transformation processes such as hydrogen production and refining. LUC land use change, BECCS bioenergy with carbon capture and storage, DACCS direct air capture and carbon storage, DORCS direct ocean removal and carbon storage, ERW enhanced rock weathering, BC black carbon, $NH_3$ ammonia, NMVOC non-methane volatile organic compounds, $NO_x$ nitrogen oxides, OC organic compounds, $SO_2$ sulfur dioxide, $GtCO_2eyr^{-1}$ gigatonnes of carbon dioxide equivalent per year, $Tgyr^{-1}$ teragrams per year.

details in Supplementary Table 2). Gross FFI $CO_2$ emissions reach a cumulative total of 240 $GtCO_2$ from 2025-2050 in the HIGH CDR scenario, compared to 137-160 $GtCO_2$ in the LOW and MODERATE CDR scenarios (Fig. 3a).

Total gross $CO_2$ removal is also depicted in Fig. 3a, highlighting only CDR approaches without accounting for $CO_2$ removal by bioenergy crops for photosynthesis. The HIGH scenario shows a rapid increase in $CO_2$ removal by 2050, with BECCS (4.6 $GtCO_2$ $yr^{-1}$), DACCS (4.2 $GtCO_2$ $yr^{-1}$), and ERW (2.2 $GtCO_2$ $yr^{-1}$) playing the most important roles, collectively reaching 11 $GtCO_2$ $yr^{-1}$ removal potential by 2050. DORCS would play a minor role, with deployment of under 2 million tonnes $CO_2$ per year ($MtCO_2$ $yr^{-1}$) by 2050 in Asia, owing to its high system and operational costs, especially when tied to desalination plants[12,47,48]. The deployment of DORCS is restricted by the demand for desalinated water, making regions with high

water desalination needs the prime beneficiaries, such as the Middle East.

Even without new climate policies, Asia is expected to witness a decline in air pollutants, except for ammonia ($NH_3$), from 2025 to 2050 (Fig. 3b). During this period, while most pollutants decrease across scenarios, $NH_3$ emissions show growth under the MODERATE and LOW CDR scenarios. This increase in $NH_3$ emissions could be attributed to the fact that while the phasing out of fossil fuels, driven by the emission reduction target, leads to a reduction in other criteria air pollutants, factors contributing to $NH_3$ emissions remain relatively unaffected by the greenhouse gas constraint. These factors include persistent fertilizer use for crop and bioenergy crop cultivation, ongoing meat and dairy production, as well as other urban processes such as waste decomposition and emissions from cooking activities. The HIGH CDR scenario, which is characterized by Asia's continued use

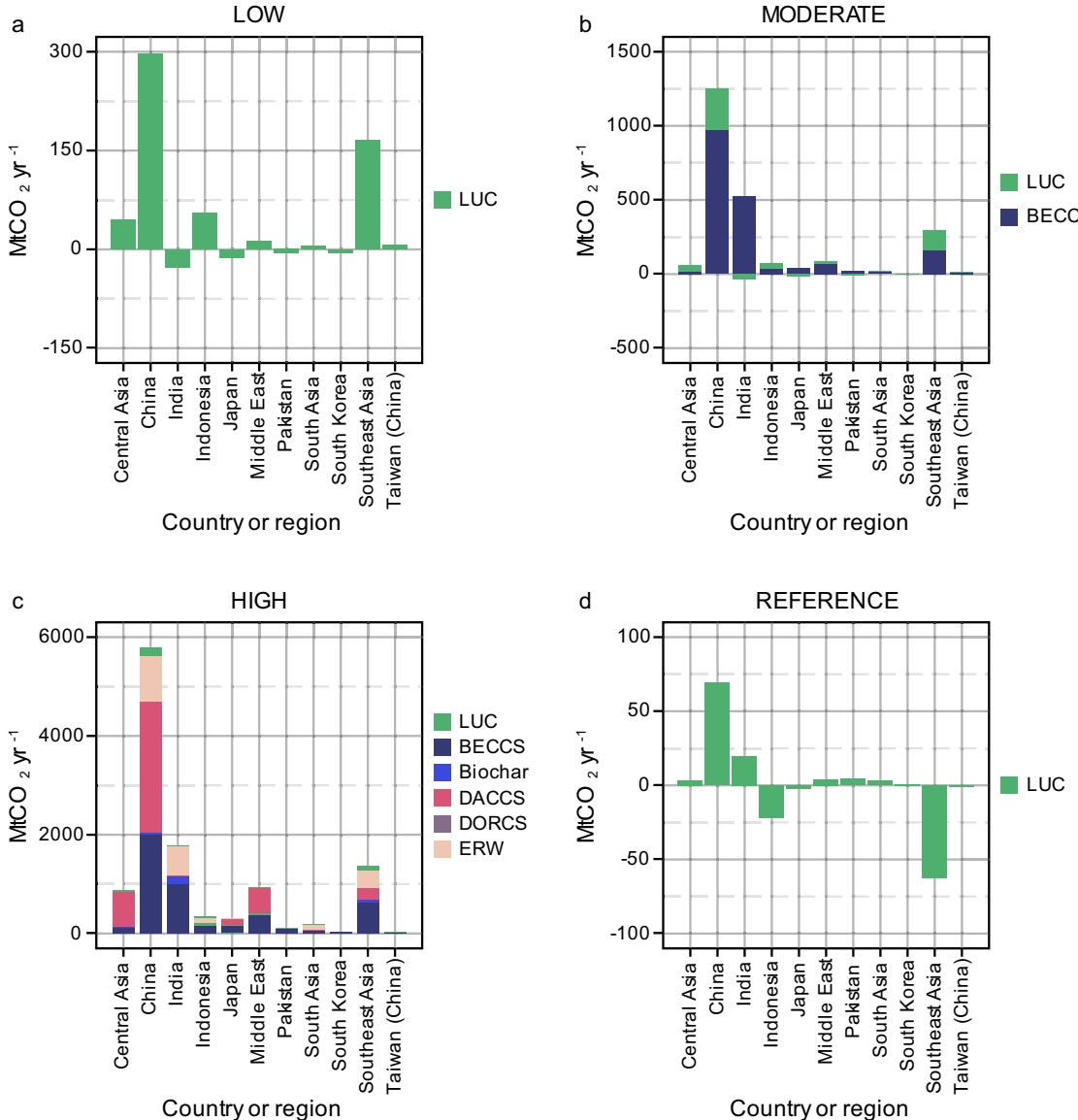

**Fig. 4 | Negative emissions type by country/region.** Carbon dioxide removal (CDR) distribution by technology/practice across Asia under various CDR deployment scenarios (**a**: LOW; **b**: MODERATE; **c**: HIGH) by 2050 alongside a no-climate-policy scenario (**d**: REFERENCE). Negative values for LUC in some countries or regions represent net positive emissions, where deforestation exceeds the afforestation rate. LUC land use change, BECCS bioenergy with carbon capture and storage, DACCS direct air capture and carbon storage, DORCS direct ocean removal and carbon storage, ERW enhanced rock weathering, $MtCO_2yr^{-1}$ million tonnes of carbon dioxide per year.

of carbon-intensive energy and inefficient practices, results in the highest total air pollutants by 2050.

Depending on resource constraints, local climate conditions, and financial capabilities, the CDR approach that countries or regions choose to invest in could vary significantly across the Asian region (Fig. 4). In the resource-rich yet fossil energy-intensive economies of China, Central Asia, and the Middle East, DACCS emerges as the most viable CDR option by mid-century. These countries can leverage DACCS to offset hard-to-abate emissions without compromising economic growth by capitalizing on their abundant fossil fuel reserves and existing carbon capture infrastructure. Conversely, countries/regions such as India, Southeast Asia, and Japan may favor BECCS. The production of biochar, a byproduct of biomass pyrolysis, plays an important role in India and Southeast Asia, complementing their BECCS strategies. ERW gains prominence in China, India, and Southeast Asia due to relatively large cropland and grassland areas. While the appeal of DORCS exists, its extremely high costs make it the least

favored option in Asia. Negative emissions from land-use change, once a key part of CDR efforts, take a backseat as novel technological solutions become more mature and cost-effective under the HIGH scenario[39].

Figure 5 illustrates the positive and negative emissions and net zero timing among Asian countries and regions in response to varying CDR reliance by 2050. Under both HIGH and LOW scenarios, four key regions—China, India, the Middle East, and Southeast Asia—emerge as the main contributors to gross positive emissions (includes bio-derived, and fossil fuel and industry), each surpassing 1 $GtCO_2yr^{-1}$ emissions by mid-century. The Middle East and South Asia are the only regions with higher emissions under the LOW CDR scenario compared to the HIGH CDR scenario, highlighting the challenges these regions face in emission reductions without novel CDR technologies (Fig. 5a–c).

To counterbalance residual emissions, the most significant $CO_2$ removal efforts would be in China, India, the Middle East, and

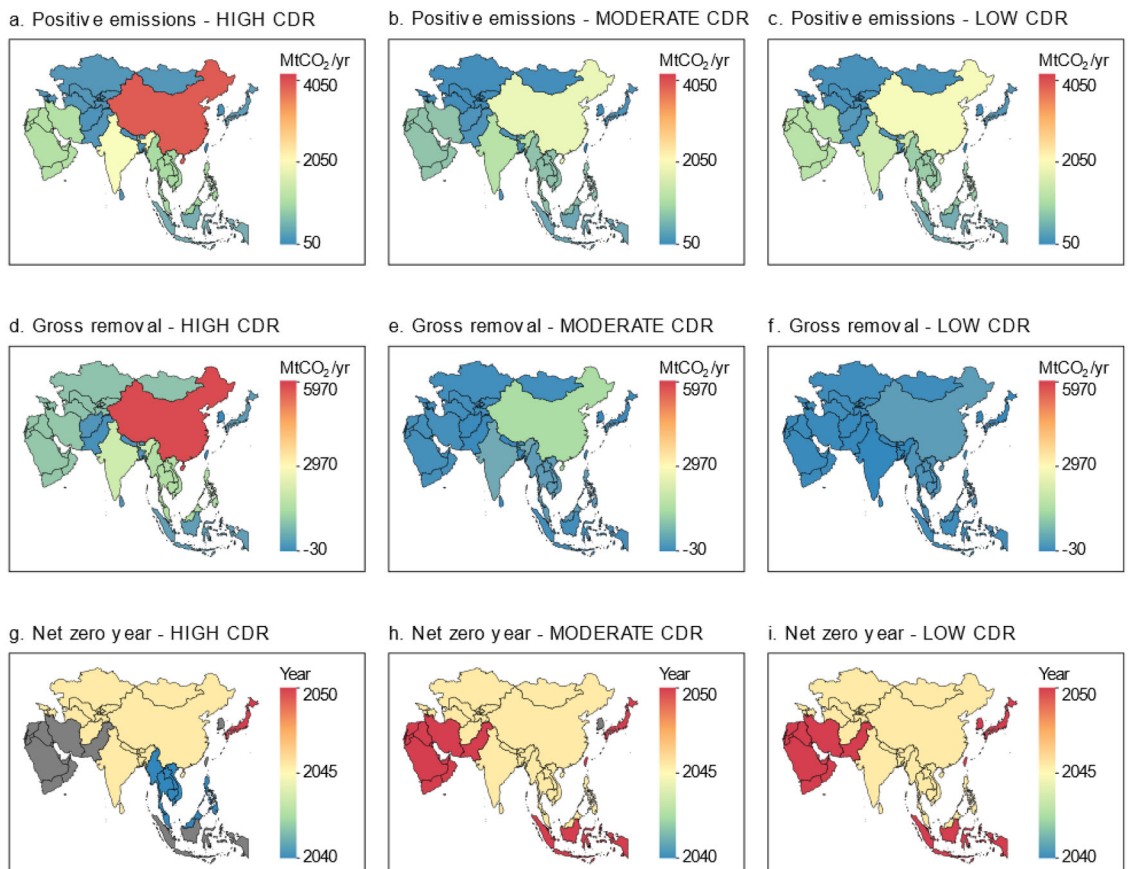

**Fig. 5 | Impact on positive and negative emissions and net zero timing.** The results include the distribution of total positive $CO_2$ emissions by 2050 (includes both fossil fuel and industry and bio-derived $CO_2$) (**a**–**c**), the distribution of total gross carbon dioxide removal (CDR) by 2050 (**d**–**f**), and the timing of domestic net zero $CO_2$ emissions (**g**–**i**). Gray color in panels **g**–**i** indicates that the country or region does not attain domestic net zero $CO_2$ before 2050. The results show that under the HIGH CDR scenario, the following countries' (or regions') annual $CO_2$ emissions would exceed 1 $GtCO_2$ per year by mid-century: China (4.1 $GtCO_2yr^{-1}$), India (2.4 $GtCO_2yr^{-1}$), the Middle East (1.2 $GtCO_2yr^{-1}$), and Southeast Asia (1.2 $GtCO_2yr^{-1}$). The same four countries or regions individually cross the 1 $GtCO_2$ per year threshold by 2050 under the LOW CDR scenario; that is, China (2.3 $GtCO_2yr^{-1}$), India (1.7 $GtCO_2yr^{-1}$), the Middle East (1.4 $GtCO_2yr^{-1}$), and Southeast Asia (1.0 $GtCO_2yr^{-1}$). Under the HIGH CDR scenario, $CO_2$ removal by mid-century would be mainly concentrated in China (6 $GtCO_2yr^{-1}$), India (1.8 $GtCO_2yr^{-1}$), and the Middle East (0.9 $GtCO_2yr^{-1}$). Under the MODERATE CDR scenario, these removal capacities in the three countries/regions significantly reduce to 1.2, 0.5, and 0.08 $GtCO_2yr^{-1}$, respectively. Also, under the HIGH CDR scenario, the assumption is that countries or regions such as Indonesia, the Middle East, Pakistan, South Korea, and Taiwan (China) may choose to pursue cheaper mitigation routes by purchasing foreign CDR instead of achieving domestic net zero before the end of 2050, which is relatively more expensive. However, under the MODERATE and LOW CDR scenarios, Asian countries or regions could advance or maintain the year of attaining domestic net-zero $CO_2$ emissions compared to the case of the HIGH CDR scenario. Our analysis considered Island countries/regions and Territories in Asia but they are not included in the map. $MtCO_2yr^{-1}$: million tonnes of carbon dioxide per year.

Southeast Asia. Under the HIGH scenario, China, India, and the Middle East are the primary contributors to $CO_2$ removal. Interestingly, the LOW CDR scenario leads to net positive values (negative sign in the figure) in the LUC sector for countries such as India (26 $MtCO_2yr^{-1}$), Japan (12 $MtCO_2yr^{-1}$), Pakistan (6 $MtCO_2yr^{-1}$), and South Korea (5 $MtCO_2yr^{-1}$), suggesting more deforestation than afforestation activities in these regions by 2050 (Fig. 5d–f). These countries will therefore require a more rapid reduction in net emissions within their fossil fuel and industry sectors to facilitate their attainment of domestic total net zero $CO_2$ under the LOW CDR scenario before 2050.

Figure 5g–i also shows when each Asian country or region is expected to achieve net-zero $CO_2$ emissions. Under the HIGH CDR scenario, Southeast Asia leads, as it reaches net-zero emissions the earliest. In contrast, several countries opt for more economical foreign CDR purchases instead of achieving domestic net zero emissions by 2050. In this context, "foreign CDR purchase" refers to countries' plans to buy CDR credits from other countries instead of achieving net-zero emissions domestically. Bhutan (South Asia), Suriname, and Panama are among the few countries already at net-negative emissions due to their high forest cover[49]. This group of 'net-negative' countries has

plans to sell forest carbon offset credits to other more polluting countries under the Paris Agreement[49,50]. For instance, to align with the Paris Agreement goals, South Korea's Long-Term Strategy (LTS) includes provisions that allow the country to purchase carbon offsets from overseas[51]. By purchasing these credits, the more polluting countries can claim to have effectively paid to reduce their emissions – a claim that most experts continue to question and raise concerns about[49]. On the other hand, under the MODERATE and LOW CDR scenarios, it becomes economically unattractive to purchase foreign CDR; hence, Asian countries and regions prioritize achieving domestic net-zero $CO_2$ before 2050.

The electricity sector is at the heart of every energy transition, and a rapid decarbonization of the sector could be a very disruptive process with huge socioeconomic implications, especially for countries in the Global South[52,53], who need time to develop under cheaper fossil fuel resources[49]. Our LOW and MODERATE CDR scenarios show even higher electrification rates from renewables and nuclear. Across the CDR pathways examined in this study, we find that the cumulative capacity of fossil fuel power plants without CCS technology, which would retire prematurely before reaching the end of their expected

operational lifetimes in Asia, amounts to 1100 gigawatt (GW) (HIGH CDR), 1260 GW (MODERATE CDR), and 1305 GW (LOW CDR) (more details in Supplementary Table 3). To provide context, the total installed capacity in Asia in 2020 was 3995 GW[54]. Due to its rapid adoption of decarbonization technologies, the absence of any novel form of negative emissions (LOW CDR scenario) throughout the modeling period would lead to a cumulative stranding of $9.5 trillion from 2015 to 2050. This is approximately twice the stranded cost compared to a pathway heavily reliant on CDR (HIGH CDR scenario). Since the HIGH CDR scenario delays the rapid phasing down of fossil fuel power plants, as well as the installation of new renewable energy sources, and nuclear technologies, the investments in these technologies are relatively lower compared to the LOW CDR scenario (more details in Supplementary Table 4). Between 2046 and 2050, the newly installed capacities and investments in these decarbonization technologies under the HIGH CDR scenario would amount to 916 GW and $2.6 trillion, respectively. In contrast, under limited reliance on CDR (LOW CDR), these figures would rise to 2322 GW and $7.6 trillion, respectively.

In all scenarios, coal power plants without CCS face the most significant risk of stranding, with the potential for cumulative stranded capacity ranging from 40% to 70% and related costs between 50% and 85%. Conversely, solar power stands out with the highest percentages in newly installed capacities, accounting for 30–35% of the total. However, when considering the cost of newly installed capacities, solar and nuclear power occupy the top positions, representing 20%–30% each. A more detailed description of our approach to estimating capital stock turnover is provided in the Supplementary Note 1.

### Impact on land allocation, water, and fertilizer demands

Ambitious climate goals in Asia will lead to land-use changes in the region, as shown in Fig. 6a. The HIGH CDR scenario would allocate more land annually to bioenergy crop cultivation than the LOW and MODERATE CDR scenarios would. Interestingly, croplands for food and non-food crop cultivation would receive slightly more allocation under HIGH CDR (more details in Supplementary Table 5).

As depicted in Fig. 6b, potential trade-offs exist for water consumption under high CDR reliance levels. With its greater CDR deployment, the HIGH CDR scenario could risk increasing water demand in sectors or for activities such as bioelectricity CCS, bioenergy crop cultivation, $CO_2$ removal, and industry. Due to the deployment of 4.6 $GtCO_2yr^{-1}$ of BECCS by mid-century under the HIGH CDR scenario compared to 1.9 $GtCO_2yr^{-1}$ in the MODERATE CDR scenario, about 3.6 cubic kilometers ($km^3$) per year of water will be consumed for bioelectricity CCS by 2050 under the former scenario compared to 1.7 $km^3yr^{-1}$ under the latter scenario. Results from the previous sections also show an increased affinity for electricity consumption in final energy under the MODERATE and LOW CDR scenarios. As such, water consumption for electricity generation would reach 76–80 $km^3yr^{-1}$ under these two scenarios by 2050 compared to 64 $km^3yr^{-1}$ under the HIGH CDR scenario. Furthermore, by 2050, about 15 $km^3yr^{-1}$ of water will be consumed by DACCS to remove $CO_2$ from the atmosphere under the HIGH CDR scenario, which is completely avoided in the MODERATE and LOW CDR scenarios.

Net zero goals also amplify fertilizer demand, particularly for bioenergy crop cultivation, as shown in Fig. 6c. While the REFERENCE scenario's total nitrogen fertilizer requirements remain comparable to net-zero pathways, its allocation to bioenergy crops is considerably lower. Furthermore, the higher land allocation for bioenergy crop cultivation under HIGH CDR scenario increases the demand for fertilizer in cultivating these bioenergy crops compared to demand under MODERATE and LOW CDR scenarios.

Figure 7 highlights land allocation towards bioenergy crops and other agro-land allocation (crops, grass, other arable, pasture, and shrubs) among Asian countries and regions in response to varying CDR

reliance from 2025 to 2050. We find that while HIGH CDR scenario has higher annual land allocation towards bioenergy crop cultivation, the change in this land use type over the next three decades among Asian countries and regions is relatively higher in LOW CDR scenario compared to HIGH and MODERATE CDR scenarios. This is because bioenergy consumption (without CCS), similar to other zero-carbon energy sources, increases under limited reliance on CDR. As such, land allocation for bioenergy crop cultivation under the LOW CDR scenario will expand significantly between 2025 and 2050 among Asian countries and regions (Fig. 7a–c). In all pathways, South Korea exhibits the highest change in land allocation for cultivating bioenergy crops at over 3000% increase under LOW CDR scenario compared to less than 1500% under HIGH CDR scenario. Taiwan (China) exhibits the lowest changes in bioenergy cropland allocation especially under MODERATE CDR scenario at a 50% increase.

Aggregated land allocation for agro-land use types such as crops, grass, shrubs, and pasture witness decreasing growth rates over the next three decades across all scenarios. The reduction rates are least severe under the MODERATE CDR pathway compared to HIGH and LOW CDR pathways. Under higher expectations for CDR (HIGH CDR scenario), the most severe change in agro-land use (excluding biomass) would be recorded in Japan at −10.5% followed by Indonesia (−8.4%), and Pakistan (−7.7%). Similarly, the least affected country/region under this pathway would be South Korea (−1.9%), followed by South Asia (−2%), and Central America (−2.6%) (Fig. 7d–f).

### Policy implications

In this study, we highlight the risks of overreliance on CDR leading to delayed decarbonization efforts in the near term. It is imperative for governments and other key stakeholders to prioritize policies that emphasize non-CDR mitigation strategies, while viewing carbon removal as a complementary strategy. Policies should be structured to prevent decarbonization and carbon removal from substituting one another. This can be achieved through policy designs that separate CDR deployment from emission reduction objectives[24,55]. Moreover, while decarbonization efforts should take precedence in achieving the 1.5 °C target, CDR will play a crucial role in ensuring the success of end-of-century targets. The cost associated with CDR poses a significant limitation, and policies should be implemented to facilitate the rapid scaling of novel CDR technologies. Furthermore, it is essential for policies that establish carbon removal targets to be grounded in realism. These technologies are still in their infancy, with numerous uncertainties surrounding their effectiveness and scalability. Betting on the potential success of large-scale carbon removal carries significant risks and could prove detrimental to the Paris Agreement target if these technologies fail to meet expectations[56]. We discuss these policies in detail as follows.

CDR moral hazard or mitigation deterrence has been widely discussed[46,57–59], and one of the key ways to pursue CDR without shifting attention from near-term deep decarbonization is to set separate CDR and emission reduction targets. Separate CDR and emission reduction targets could be a valuable strategy to prevent potential delays in decarbonization, as they would discourage reliance on unproven future CDR deployment[24]. Setting distinct CDR targets allows explicit focus on ramping up deployment and investment in negative emissions technologies and nature-based solutions, which facilitates scaling up CDR to levels required to achieve net-zero/net-negative emissions[25]. In an environment in which emission reduction policies are independent of negative emissions, one cannot replace the other. Negative emissions would be available only for the specified amount required without delaying emission reduction[24,25,60]. For example, Australia, New Zealand, and the United Kingdom (UK) treat nature-based CDR interchangeably with emissions reductions, with no limit on how much CDR can count toward overall climate targets. Sweden takes a different approach, setting

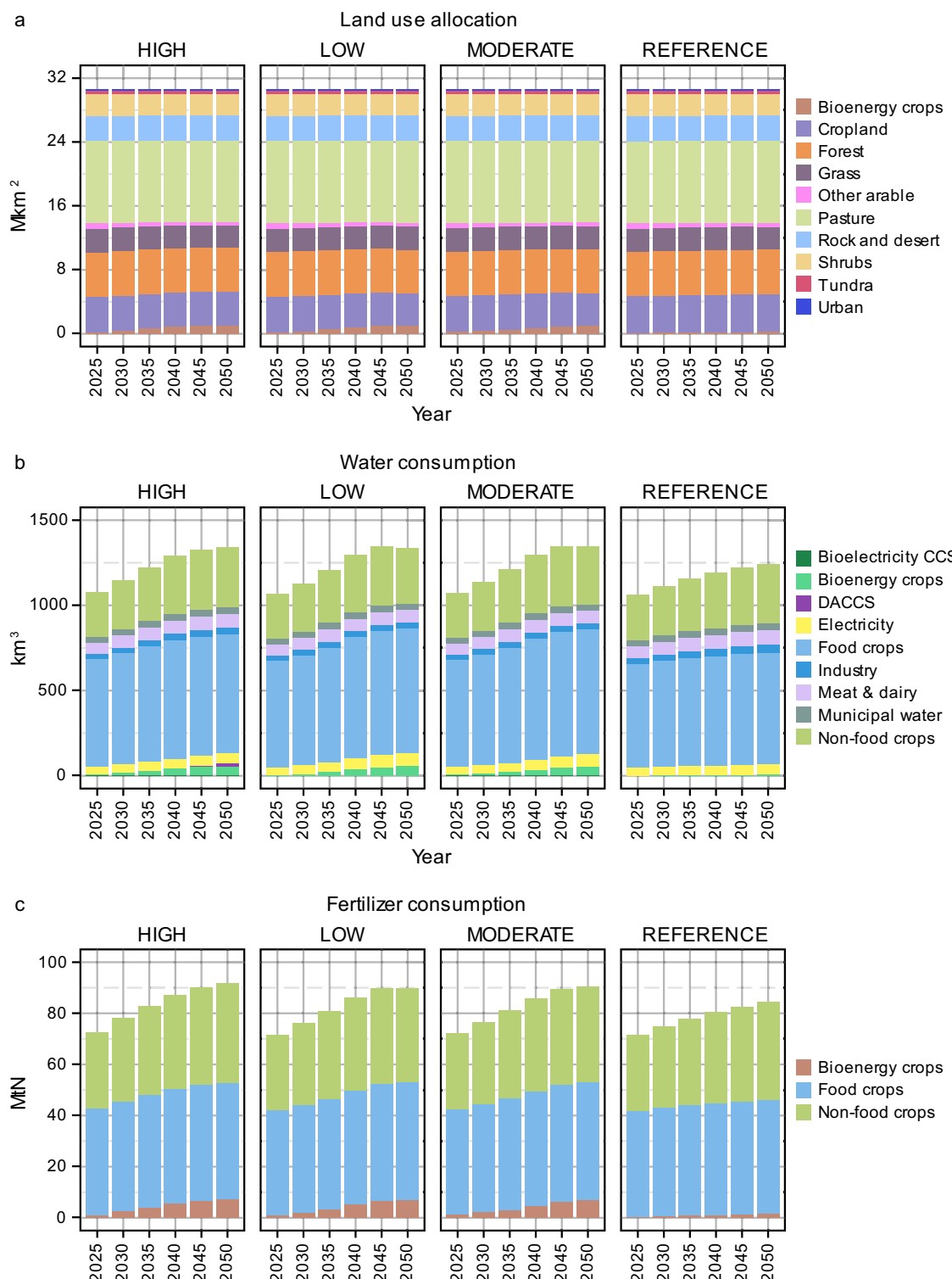

**Fig. 6 | Impact on land, water, and fertilizer use.** Asia's aggregated land use allocation (**a**). Water consumption by sector in Asia (**b**). Fertilizer demand by crop source in Asia (**c**). All results are from 2025-2050 for different levels of carbon dioxide removal (CDR) reliance (HIGH, MODERATE, and LOW) in Asia alongside a no-climate-policy scenario (REFERENCE). CCS carbon capture and storage, DACCS direct air capture and carbon storage, Mkm² million square kilometers, km³ cubic kilometers, MtN million tonnes of nitrogen.

separate emission reduction and CDR targets on its path to net-zero emissions by 2040[61]. With separate targets, the role of CDR shifts from primarily offsetting residual emissions to more of a complementary mitigation measure[62]. In other words, CDR deployment is not contingent on high fossil fuel use and emissions. Distinct targets

enable greater transparency in tracking progress on emission reductions versus negative emissions[24]. This allows for course correction if efforts lag in either domain.

Our findings highlight the decreasing role of land use in offsetting emissions towards mid-century, stressing the importance of making

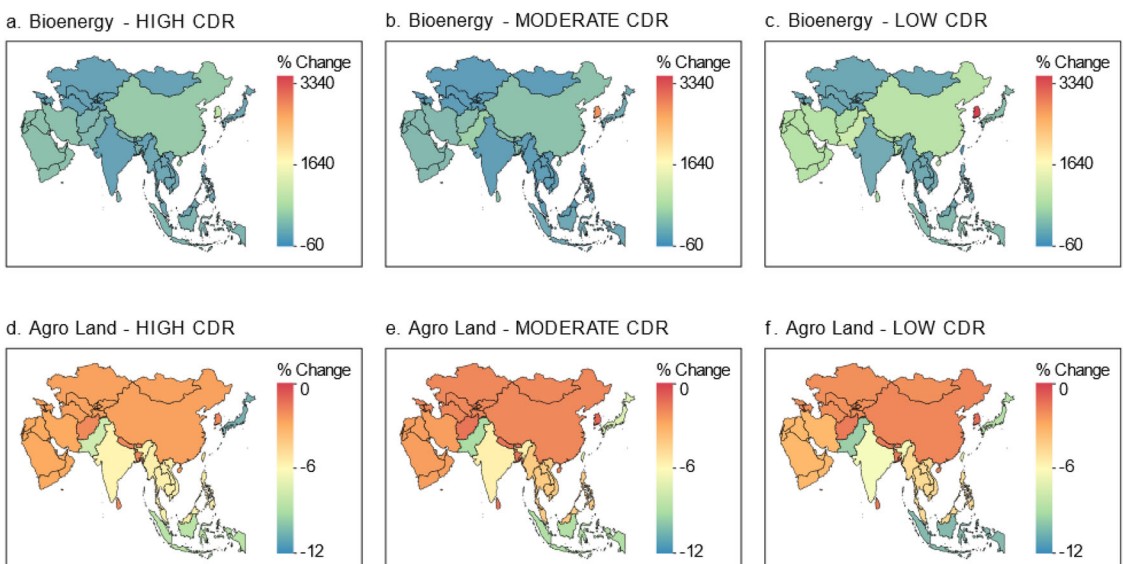

**Fig. 7 | Impact on land allocation towards bioenergy crops and other agro-land allocation.** The results include the percent change in land allocation to bioenergy crops between 2025–2050 (**a**–**c**) and the percent change in other agro-land allocation (crops, grass, other arable, pasture, and shrubs) between 2025-2050 (**d**–**f**). CDR carbon dioxide removal. Our analysis considered Island countries/regions and Territories in Asia but they are not included in the map.

room for the development and deployment of novel forms of negative emissions. Afforestation and reforestation projects are susceptible to pests, natural hazards, and extreme weather events[63,64], limiting their effectiveness as long-term CDR solutions. In conjunction with afforestation and reforestation, the adoption of novel CDR technologies such as BECCS and DACCS could help offset residual emissions. However, it is essential to note that these novel CDR approaches are associated with high costs, which may hinder their development and deployment trajectory. Governments can play a pivotal role in incentivizing novel CDR deployment, as exemplified by the Inflation Reduction Act (IRA) of 2022. This legislation expanded the 45Q tax credit for direct air carbon capture by 360% (from $50 to $180 per ton)[65].

While CDR technologies are required to achieve the 1.5 °C target, it is crucial to establish realistic targets for carbon removal. Conservative estimates suggest that novel CDR technologies must reach approximately 4.2 (with a range of 3.7–6.2) gigatonnes of $CO_2$ per year by 2050 to remain on track for the 1.5 °C target[66]. However, the current deployment of these novel CDR technologies stands at only 0.002 gigatonnes of $CO_2$ per year[3]. This means that these technologies will need to increase by approximately 1800-fold between now and around 2050. To put this into perspective, the share of renewables in total energy supply from now to 2050 to achieve net zero emissions would need to increase by 5-fold, while final energy demand from hydrogen would need to increase by 3-fold between 2030 and 2050[67]. Betting on uncertain upscaling of CDR may lead to high-temperature overshoot, with potentially irreversible climate impacts[56] if the expected multi-gigatonne scale CDR fails to materialize. Therefore, policymakers must approach carbon removal targets with caution and ensure that they are based on realistic assessments of technological capabilities and their sustainability issues. Given our relatively greater experience with decarbonization measures, policies should primarily focus on initiatives such as renewables, electrification, energy efficiency improvement, and carbon-neutral fuels to account for 80–90% of the necessary reduction in net emissions. CDR will then play a crucial role in offsetting the remaining 10–20% of residual emissions, particularly from hard-to-abate sectors.

## Discussion

In this study, we set out to quantify the potential impacts of varying reliance on CDR for reaching net-zero GHG emissions in Asia by mid-century. Our findings indicate that over-dependence on CDR deployment poses notable risks across regional energy-land-water system. Expecting multi-gigatonne of CDR by mid-century leads to continued carbon lock-in, which hinders rapid transitions to renewable energy systems, electrified transport, highly efficient buildings, and other cleaner solutions. Residual emissions remain higher due to a slower phase down of fossil fuels and inefficient practices. Furthermore, high reliance on CDR makes achieving domestic net zero economically expensive for some Asian countries and regions. As a result, such countries or regions may pursue purchasing foreign CDR instead of achieving domestic net zero emissions before the end of 2050. However, with lowered reliance on CDR, achieving domestic net zero becomes more economically attractive than purchasing foreign CDR, allowing all Asian countries or regions to attain domestic net zero $CO_2$ emissions before the end of 2050. There could also be significant co-benefits for air pollutants when CDR reliance is minimized. Furthermore, high CDR reliance tends to increase annual bioenergy cropland allocation, reduce agro-land allocation (excluding biomass), and increase water consumption and fertilizer demands.

Nonetheless, some reliance on CDR is indispensable, as immediate cuts and decarbonization alone cannot achieve net-zero emissions across all sectors, particularly in industrial processes. Novel techniques, such as DACCS and BECCS, can serve critical functions when deployed responsibly. Conventional natural solutions, such as afforestation/reforestation, remain essential in the near term[39] and should be prioritized where feasible before scaling up emerging and novel options. With the Paris Agreement's 1.5 °C remaining carbon budget shrinking rapidly, the message is clear – the world must make decarbonization the prime focus, but should not underestimate CDR's complementary role. Near-term efforts must center on ambitious policies and targets that minimize fossil fuel use and emissions across all sectors[68]. In the mid-to-long-term, moderate CDR integration can help offset residual emissions in selected applications while avoiding weakening mitigation incentives. However, reliance levels must align with ethical principles[69,70] and development goals.

## Methods

### Rationale for focusing on Asia

Here, we explain our rationale for focusing on Asia in the current work. With over 4.75 billion people (in 2023)[71], Asia accounts for about 50% of global primary energy consumption and over 50% of global GHG emissions today. Without new climate policies beyond those already existing (REFERENCE scenario), the region's global primary fossil fuel consumption and GHG emissions could reach approximately 470 EJ and 40 $GtCO_2eyr^{-1}$, respectively, by mid-century. As the world aims to limit global warming to below 1.5 °C by 2100, insufficient climate ambition from Asia could render even sufficient ambition from the rest of the world ineffective. Therefore, Asia is poised to play a crucial role in the dynamics of global climate change mitigation, consistent with the Paris Agreement.

Asian countries/regions such as China, Japan, South Korea, and India have already announced net-zero emission deadlines by 2050-2070. However, these ambitions would not be sufficient to achieve a 1.5 °C target by 2100 if the rest of the world were to follow similar levels of ambition[72,73]. Existing climate ambitions for Asia and the rest of the world would, therefore, need to be ratcheted to increase our chances of achieving a temperature change of below 1.5 °C by 2100[74]. In addition, without equity considerations, the largest share of global future CDR deployment[7,12] and all potential impacts that come with it could be mainly concentrated in Asia.

### Scenario formulation

The central pathway in this study follows a stylized path where we target net zero GHG emissions in Asia. Consistent with 1.5 °C, we modeled an emission reduction pathway where 2030 GHG emissions in Asia are approximately 45% lower than the levels in 2019[2]. Under this pathway, GHG emissions in Asia peak before 2025 and reach net zero around mid-century. In this work, a net GHG emissions pathway has been modeled over the conventional net zero $CO_2$ pathway in most regional and global studies. This is due to the urgent need for non-$CO_2$ GHGs to be concurrently reduced alongside $CO_2$. Mitigating non-$CO_2$ emissions in the short term, up to 2030, can potentially lower the peak temperature increase throughout this century, thereby reducing the risk of exceeding the 1.5 °C threshold. Over the long term, the quantity of residual non-$CO_2$ emissions will ultimately impact the temperature stabilization level when global $CO_2$ emissions achieve a net-zero state[4].

Following the central pathway are three different levels of CDR reliance, modeled based on the number of CDR approaches available for deployment and constraints on how much $CO_2$ could be removed at any point in time (Table 1).

In the scenario where the reliance on CDR is labeled as HIGH, six CDR approaches are deployed from 2025 onwards. These CDR approaches include AR, Biochar, BECCS, DACCS, DORCS, and ERW, and they are deployed without constraining how much $CO_2$ they can remove from the atmosphere per year. With no limit on negative emissions, gross CDR deployment in Asia endogenously increases to about 12 $GtCO_2yr^{-1}$ by 2050 in this scenario.

The obstruction of mitigation efforts when relying heavily on CDR is by design integrated into climate modeling[18]. CDR unavoidably displaces some of the immediate mitigation actions. Since scenarios in Integrated Assessment Models (IAMs) like GCAM are designed to minimize mitigation costs over the century, the inclusion of CDR alters the distribution of these costs throughout that time-frame. The introduction of CDR into IAMs results in higher emissions in the near term compared to scenarios with limited or without novel CDR[30,44]. This increase implies that immediate climate action becomes less stringent and, consequently, less costly. This is not because CDR makes near-term mitigation more expensive, but rather because the availability of CDR reduces the overall cost of mitigation over the entire century[13,18,43].

Rogelj et al. proposed a framework that provides a logic to separate emission reduction from CDR deployment[55]. This framework enables studies to explore future CDR deployment as an independent variation under a desired temperature outcome. In that case, in an environment where emission reduction policies are independent of negative emissions, one cannot replace the other. Negative emissions would be available only for the specified amount required without delaying emission reduction[24,25]. Based on these principles, we developed two additional scenarios, MODERATE and LOW, each with distinct CDR targets in terms of available CDR options and removal capacity. The descriptions of these two scenarios are provided below.

In the scenario where the reliance on CDR is labeled as MODER-ATE, only AR and BECCS are deployed. AR is endogenously deployed, and the rationale for choosing BECCS as the sole novel CDR over other alternatives is that BECCS is the most represented novel CDR in IAM-based studies and has relatively low cost and higher technological readiness level. BECCS can also provide low-carbon energy in the electricity, refinery, and hydrogen sectors as it captures $CO_2$. In this scenario, CDR has been deployed with a specified upper limit on how much $CO_2$ they can remove per year. The deployment of BECCS and the upper limit on gross negative emissions begin in 2025 and continues until the end of 2050. Here, global BECCS deployment is exogenously limited to 2 $GtCO_2yr^{-1}$. 2 $GtCO_2yr^{-1}$ removal here is to represent the current gross CDR available globally[3]. Asia's share of this BECCS amount is endogenously deployed, reaching an annual average of 1.8 $GtCO_2yr^{-1}$ from 2025 to 2050. In this scenario, total gross $CO_2$ removal in Asia reaches 2.3 $GtCO_2yr^{-1}$ by 2050.

In the LOW scenario, only AR is deployed for negative emissions. Here, we limit CDR by novel means to 0 $GtCO_2yr^{-1}$ throughout the modeling period. AR is endogenously deployed, where total gross $CO_2$ removal reaches 0.5 $GtCO_2yr^{-1}$ by 2050. The assumption here is that novel CDR would fail to materialize in Asia before mid-century. Besides, several Asian countries, including China, have yet to include quantified negative emissions in their long-term national climate strategies[75].

Several IAMs primarily focus on AR and BECCS for modeling negative emissions. Only a few IAMs include DACCS. AR, BECCS, and DACCS are the only three CDR approaches in the publicly released version of the Global Change Assessment Model (GCAM). To achieve our objectives, we modified the core version of GCAM5.4 to create a version called "GCAM-TJU." This modified version includes three additional novel CDR methods: biochar, ERW, and DORCS. To our knowledge, only a handful of studies[12,30,31,44,62,76] have gone beyond BECCS-AR-DACCS from an IAM perspective. The inclusion of a wide array of CDR approaches (i.e., BECCS-DACCS-AR-DORCS-Biochar-ERW) here is in an attempt to offer a different perspective on diverse CDR approaches compared to what has been done in previous studies. Detailed descriptions and assumptions for all six CDR approaches are provided in the Supplementary Note 2 to Supplementary Note 7.

The shared socioeconomic pathways (SSPs) are part of a framework developed by climate scientists to describe different future worlds based on varying socioeconomic conditions. The SSPs outline five broad narratives for future societal development, each with different challenges for mitigation and adaptation to climate change[77–80]. The SSP narratives provide context for quantitative modeling and scenario exercises to explore climate policy options. In this work, all scenarios are modeled under the SSP2 baseline (where social, economic, and technological trends follow historical patterns), which is considered a central 'best-estimate' case[81].

Additionally, a comparison of our results with those from previous Asia-based IAM studies is discussed in detail in Supplementary Discussion 4. We have also included a sensitivity analysis to validate the robustness of our initial findings, with the approach and results detailed in Supplementary Discussion 5.

## Model description

The Global Change Analysis Model (GCAM) is an integrated assessment model that represents interactions between five systems: economy, energy, agriculture, land use, and climate. GCAM divides the world into 32 geopolitical regions and further divides land areas into 384 subregions and water resources into 235 basins[82]. The model operates in 5-year time steps from 2015 to 2100, solving for equilibrium prices and quantities in energy, agricultural, land use, water, and greenhouse gas markets in each time period and region. As a recursive dynamic model, GCAM's solutions for each period depend only on conditions in the last modeling period. GCAM's outputs are driven by exogenous assumptions about population, labor participation, productivity, resources, technologies, and policies in each region[83]. The model tracks emissions of 24 different gases, including greenhouse gases, short-lived species, and ozone precursors, endogenously based on the resulting energy, agriculture, and land-use systems[74]. A more detailed description of the model is provided in Supplementary Method 1 to Supplementary Method 5 alongside its land (Supplementary Note 8), water (Supplementary Note 9), and fertilizer (Supplementary Note 10) modules.

## Data availability

All modeling data for this study are available in a public repository accessible at https://doi.org/10.5281/zenodo.11254051.

## Code availability

GCAM is an open-source community model available at https://github.com/JGCRI/gcam-core/releases. The particular version of GCAM and additional input files associated with this study are available at https://doi.org/10.5281/zenodo.11254051.

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

## Acknowledgements

We are grateful to the National Natural Science Foundation of China for supporting the current study with funding numbers 52176125 & T2341001 received by H.L.; H.M. was supported by the National Research Foundation of Korea grant RS-2023-00219466.

## Author contributions

J.D.A., H.L., C.J., M.Y., H.A., and H.M. conceived and designed the research. J.D.A. developed the scenarios, conducted the modeling, and wrote the first draft of the paper. J.F. and H.M. contributed to the modeling tools. S.A. and H.A. co-led parts of the assessment and contributed to data analysis. H.L., C.J., M.Y., H.M., and D.T.H. supervised the research. J.D.A., C.J., H.L., M.Y., S.A., J.F., H.A., D.T.H., and H.M. contributed to the writing of the article.

## Competing interests

The authors declare no competing interests.
