## [Peer Review File · Nature Communications]

REVIEWER COMMENTS

Reviewer #1 (Remarks to the Author):

The paper models three scenarios of CDR reliance with the model GCAM to investigate the impacts of high reliance on CDR in mitigation pathways on water, land and energy. The study includes six different CDR techniques, though the focus is not on comparing the techniques but assessing the impacts of an unconstrained reliance on the whole portfolio of CDR. A first main finding is a quantification of the fossil lock-in that comes with high CDR reliance. High reliance on CDR displaces renewable energies, electrification, and demand reduction in IAM scenarios and leads to continued fossil emissions in comparison to low CDR scenarios. A second main finding is a quantification of the impacts of high CDR reliance. High CDR reliance shows significant adverse impacts, for instance using a majority of the cropland in some countries of the region due to Biochar development and thereby threatening food security.

The research question of the paper addresses an important aspect of mitigation pathways. By including a larger portfolio of CDR techniques, the study addresses a serious limitation of many existing CDR studies and promises to provide a more realistic vision of what reliance on negative emissions in reaching Net-zero would involve. The study design is suitable to investigate this question. The chosen scope of Asia offers insightful scenarios.

The paper would benefit if it addressed some of the issues below:

- The central policy conclusion of separating CDR deployment from emissions reductions is well supported by the results, though an open question regarding its robustness remains when reading the paper. One question is to what degree the results depend on assumptions relating to the different CDR techniques. Especially the reliance on Biochar seems to have a large influence on key results, for instance, using almost all (?) cropland of the region. If so, this seems like an unacceptable scenario in this regard. While this supports the main conclusion of the study, the study would profit from increasing the robustness of this finding by modeling additional scenarios. (One could replace Section 5 in my opinion to do so.) Relevant questions are: How would constraints relating to the SDG (for instance resulting from food security) alter the CDR portfolio used by the models? How would this impact results relating to land use and water demand? Moreover, how robust are these findings in relation to assumptions on how biochar is represented (as it drives these results)? Including sensitivity analysis (also in relation to some model settings, see below) would significantly strengthen the conclusion that high CDR reliance has severe adverse impacts.

- One major finding is that high CDR reliance involves mitigation deterrence. This is a well-known finding in the literature and should be put into perspective (cf. Lenzi 2018). Mitigation deterrence is a logical implication of IAMs that optimize for whole century costs. To address this shortcoming, Rogelj et al. (2021) propose a scenario design to detach CDR deployment from emissions reductions. Applying this logic could make results more policy-relevant by providing policymakers with different options for specifying Net-Zero targets. As this is not the focus of the study, one might simply discuss the conclusions. The results are still relevant, as they quantify deterrence under a set of assumptions.
- The paper would profit from reporting the discount rate and making transparent, in how far key findings are dependent on the chosen SDR. In the context of modeling CDR, cost-minimization and high discount rates are known to have a large impact on key results, such as mitigation costs, net-zero timing and overall deployment of negative emissions (cf. Emmerling et al. 2019; Grant et al. 2021; Riahi et al. 2021). The discount rate is not discussed in the current manuscript, though the authors refer to Fuhrmann et al. 2021 for discussions. Transparency of the results could be improved by including a short discussion. Varying the discount rate could alter central results such as the high demand for cropland and the significantly higher costs in Moderate and Low. However, this issue is not particular for this study but reflects a lack of transparency of integrated modeling overall (.
- Readability of Fig. 7 might be improved by using absolute numbers for a-c given the low starting values.

Literature

Emmerling, J., Drouet, L., van der Wijst, K.-I., van Vuuren, D., Bosetti, V., & Tavoni, M. (2019). The role of the discount rate for emission pathways and negative emissions. *Environmental Research Letters*, 14(10), 104008. <https://doi.org/10.1088/1748-9326/ab3cc9>

Grant, N., Hawkes, A., Mittal, S., & Gambhir, A. (2021). Confronting mitigation deterrence in low-carbon scenarios. *Environmental Research Letters*, 16(6), 64099. <https://doi.org/10.1088/1748-9326/ac0749>

Lenzi, D. (2018). The ethics of negative emissions. *Global Sustainability*, 1(E7), 1-8. <https://doi.org/10.1017/SUS.2018.5>

Riahi, K., Bertram, C., Huppmann, D., Rogelj, J., Bosetti, V., Cabardos, A.-M., Deppermann, A., Drouet, L., Frank, S., Fricko, O., Fujimori, S., Harmsen, M., Hasegawa, T., Krey, V., Luderer, G., Paroussos, L., Schaeffer, R., Weitzel, M., van der Zwaan, B., . . . Zakeri, B. (2021). Cost and

attainability of meeting stringent climate targets without overshoot. *Nature Climate Change*, 11(12), 1063–1069. <https://doi.org/10.1038/s41558-021-01215-2>

Rogelj, J., Huppmann, D., Krey, V., Riahi, K., Clarke, L., Gidden, M., Nicholls, Z., & Meinshausen, M. (2019). A new scenario logic for the Paris Agreement long-term temperature goal. *Nature*, 573(7774), 357–363. <https://doi.org/10.1038/s41586-019-1541-4>

Reviewer #2 (Remarks to the Author):

Carbon dioxide removal (CDR) is essential to achieve the net-zero emissions, and the deployment level of CDR technology is greatly depend on the time of net-zero emissions, all greenhouse gas emissions mitigation technologies, social and economic development, related resource constraints and etc. This paper focuses on the impacts of different CDR deployment level using GCAM model, and provides more perspective to consider CDR development. Although the topic is interesting, I think there are some issues with this paper, the authors are requested to go through the following comments.

Comments:

(1) My biggest concern is that the conclusion of this article is very obvious. By 2050, net-zero greenhouse gas emissions will be achieved. Under HIGH scenarios, greenhouse gas emissions will inevitably be higher, and fossil energy utilization will also be more. This is closely related to the scenario setting, thus the results in Table 2 are very obvious. The relationship between CDR and fossil energy consumption is consistent with other literature, and there are no new findings. Furthermore, the subsequent research results on land and water are also evident.

Overall, the analysis of this paper is not at a high level, only emphasizing the adverse impacts of CDR technology.

(2) The application of any technology will have some impacts. I believe that for achieving the specific temperature control goal of 1.5 °C, which can avoid climate loss, CDR is one of the optimal technology combinations under many constraints such as technological availability and economy factors. Therefore, this article should explain and discuss what the constraints of CDR technology are considered in the model. For example, in the high scenario of 12GtCO₂, whether this setting takes into account resource potential and other related constraints, and whether it can be achieved.

(3) The introduction of the model is very unclear, and it is unclear how the author improved the GCAM model to obtain the research results in the paper. There is a lack of explanation for the relevant models of land, water, and fertilizer.

(4) In the high scenario, there are six CDR technologies required, some of CDR technology are expensive, then why is the carbon price \$248/tCO₂ in Table 2, which is the lowest among HIGH, MODERATE, and LOW scenarios. I have some doubts about this carbon price.

(5) The scenario settings in the model lack explanation, such as reaching 12GtCO₂ by 2050 under HIGH scenario; In the MODERATE scenario, the annual average is 1.8 GtCO₂, while in the LOW scenario it is 0.5 GtCO₂ by 2050. The author needs further explanation on why this is set.

(6) The figures in the paper are not clear.

(7) The paper lacks some references, such as line64-72.

(8) There are multiple statements in the paper regarding over reliance on CDR. How is the definition of over reliance defined here, whether it is the proportion of emissions reduction exceeding or how it is defined.

(9) Policy implications are not in-depth enough. For example, non-CDR strategies as renewable ... are too general, authors should clarify specific measures.

Reviewer #3 (Remarks to the Author):

This paper models future decarbonization scenarios that rely heavily on NETs/CDR. The work is of contemporary interest and is scientifically novel, but this version of the manuscript has flaws which need to be fixed prior to publication.

1) Specific assumptions used for the NETs covered in the work are unclear. For example, it is unclear if environmental co-benefits are included in the calculations. These should be specified for full transparency.

2) The results of the work should be compared with prior work, and any discrepancies should be explained.

3) In general, the figures are of poor quality and need to be improved for easier reading.

4) A few minor (typographical/grammatical) errors in the text should be corrected.

Reviewer 1

The paper models three scenarios of CDR reliance with the model GCAM to investigate the impacts of high reliance on CDR in mitigation pathways on water, land and energy. The study includes six different CDR techniques, though the focus is not on comparing the techniques but assessing the impacts of an unconstrained reliance on the whole portfolio of CDR. A first main finding is a quantification of the fossil lock-in that comes with high CDR reliance. High reliance on CDR displaces renewable energies, electrification, and demand reduction in IAM scenarios and leads to continued fossil emissions in comparison to low CDR scenarios. A second main finding is a quantification of the impacts of high CDR reliance. High CDR reliance shows significant adverse impacts, for instance using a majority of the cropland in some countries of the region due to Biochar development and thereby threatening food security.

The research question of the paper addresses an important aspect of mitigation pathways. By including a larger portfolio of CDR techniques, the study addresses a serious limitation of many existing CDR studies and promises to provide a more realistic vision of what reliance on negative emissions in reaching Net-zero would involve. The study design is suitable to investigate this question. The chosen scope of Asia offers insightful scenarios.

The authors thank the Reviewer for their careful attention to our paper and the constructive feedback. We greatly appreciate you highlighting areas where we can strengthen our study design and writing. We have addressed each of your concerns in detail, and we hope that the revisions we have made meet your expectations. Please let us know if you have any other suggestions based on the revised version.

(1) The central policy conclusion of separating CDR deployment from emissions reductions is well supported by the results, though an open question regarding its robustness remains when reading

the paper. One question is to what degree the results depend on assumptions relating to the different CDR techniques. Especially the reliance on Biochar seems to have a large influence on key results, for instance, using almost all cropland of the region. If so, this seems like an unacceptable scenario in this regard. While this supports the main conclusion of the study, the study would profit from increasing the robustness of this finding by modeling additional scenarios. (One could replace Section 5 in my opinion to do so.) Relevant questions are: How would constraints relating to the SDG (for instance resulting from food security) alter the CDR portfolio used by the models? How would this impact results relating to land use and water demand? Moreover, how robust are these findings in relation to assumptions on how biochar is represented (as it drives these results)? Including sensitivity analysis (also in relation to some model settings, see below) would significantly strengthen the conclusion that high CDR reliance has severe adverse impacts.

We appreciate the Reviewer's concern regarding the robustness of our results. This is a critical point, as it helps us assess the strength of our study's conclusions and identify which modeling parameters have the most significant influence on the initial results. To address this concern, we have incorporated 12 new scenarios into our analysis. Some of these scenarios examine the impact of biochar modeling parameters, such as application and sequestration rates. Others explore how changes in constraints on land use for climate mitigation can affect the initial results, particularly in terms of food security resulting from cropland expansion or reduction. We have evaluated nine different impacts, spanning cropland allocation, water consumption, demands for fertilizer and biomass, electrification, renewables and nuclear deployment, mitigation cost, greenhouse gas (GHG) emissions, and negative emissions. These impacts intersect with various Sustainable Development Goals (SDGs), including goals 2, 3, 6, 7, 11, 12, and 13.

Our findings indicate that the biochar application rate and sequestration rate have a noticeable impact on cropland allocation but do not significantly affect the other indicators. Notably,

the negative emissions budget, set at 0.25-1% of GDP, does not exert a significant influence on any of the indicators. However, it is crucial to emphasize the pivotal role of the land use sector, as it has a substantial impact on several of the indicators.

Below, we have included excerpts from the manuscript for your review which can be found in discussion under “robustness check” (Kindly see discussion under “sensitivity analysis” for how the sensitivity analysis was carried out).

To assess the robustness of our initial findings, we conducted a sensitivity analysis on key modeling parameters. Our analysis indicates that, for the most part, the biochar application rate (Fig. 7), biochar sequestration rate (Supplementary Fig. 2), and the allocation of GDP towards negative emissions (Supplementary Fig. 3) have minimal impact on our initial results. However, cropland allocation is highly sensitive to changes in the biochar application rate. By 2050, in a scenario with a biochar application rate of 10t/ha, cropland allocation reaches 1.1 Mkm² compared to 1.3 Mkm² and 2.5 Mkm² in scenarios with application rates of 20t/ha and 100t/ha, respectively (Fig. 7a). Similarly, cropland allocation is strongly influenced by changes in the carbon sequestration rate of biochar. Higher removal rates lead to increased biochar cropland allocation, which in turn reduces land allocation to croplands (i.e., cropland allocation without biochar). For example, at a removal rate of 90%, biochar demand reaches 4365 Mt, compared to 4200 Mt and 4000 Mt at sequestration rates of 70% and 50%, respectively. Consequently, cropland allocation by 2050 in a scenario with a biochar sequestration rate of 90% would reach 1.1 Mkm² compared to 1.3 Mkm² and 1.5 Mkm² in scenarios with sequestration rates of 70% and 50%, respectively (Supplementary Fig. 2a).

The role of negative emissions from land-use change has the most significant impact on our initial results (Fig. 8). In our main scenarios, a carbon price that linearly increases from 10% to

100% from 2025 to 2100 was applied to the land use sector (See Method). Within this range, the percentage share of the carbon price allocated to the land use sector by 2050 reaches 40%. To assess the robustness of our initial findings involvement of land use sector in climate mitigation, we modeled two additional carbon price scenarios for the land use sector: a constant carbon price of 10% and 30% from 2025 to 2050. In the 10% carbon price scenario, land use change plays a relatively minor role as a negative emission source compared to the 30% and 40% carbon price scenarios.

In the 10% carbon price scenario (Fig. 8a), cropland allocation remains the highest at 4.14 Mkm² compared to 4.12 Mkm² at 30% and 4.10 Mkm² at 40% carbon prices. Since land use change has a limited role in offsetting residual emissions at a 10% carbon price, electrification, energy extraction, and biomass consumption would need to increase to compensate. Consequently, a 10% constant carbon price in the land use sector would lead to higher water consumption for negative emissions and energy generation compared to consumption at 30% and 40% (Fig. 8b). Additionally, the increased biomass consumption and cropland allocation in the 10% carbon price scenario would result in higher demands for total nitrogen fertilizer (Fig. 8c).

Biomass, as a primary energy source, would need to increase significantly to compensate for the lower negative emissions from land use change under a 10% constant carbon price (Fig. 8d). As a result of its substantial biomass demand, the consumption of renewables and nuclear energy tends to decrease slightly at a 10% carbon price compared with higher carbon prices in the land use sector (Fig. 8e). Furthermore, a less stringent climate pathway in a lower carbon price in the land use sector would lead to the highest residual GHG emissions (Fig. 8g), lower gross CDR allocation (Fig. 8h), and lower marginal cost (Fig. 8i).

Fig. 7 Robustness check: influence of biochar application rate on key modeling results related to regional sustainable development goals. Since biochar, as a CDR approach, is only considered in our 'High' scenario, the sensitivity analysis was exclusively conducted on this particular scenario. Water demand represents water consumption for negative emissions and energy generation.

Supplementary Fig. 2 Robustness check: influence of biochar carbon sequestration rate on key modeling results related to regional sustainable development goals. Since biochar, as a CDR approach, is only considered in our 'High' scenario, the sensitivity analysis was exclusively conducted on this particular scenario. Water demand represents water consumption for negative emissions and energy generation.

Supplementary Fig. 3 Robustness check: influence of negative emissions budget (CDR budget) on key modeling results related to regional sustainable development goals. The sensitivity analysis here is based on our 'Moderate' scenario, since this scenario represents moderate challenges and regional impacts between decarbonization and carbon removal.

Water demand represents water consumption for negative emissions and energy generation. While the negative emissions budget does yield somewhat different results across scenarios, the differences between these scenarios are relatively negligible. Thus, the line charts representing these scenarios tend to overlap due to their similarity in

outcomes.

Fig. 8 Robustness check: influence of land use sector involvement in climate mitigation on key modeling results related to regional sustainable development goals. The sensitivity analysis here is based on our MODERATE scenario, since this scenario represents moderate challenges and regional impacts between decarbonization and carbon removal. Water demand represents water consumption for negative emissions and energy generation. 'LUC' means 'land use change'.

(2) One major finding is that high CDR reliance involves mitigation deterrence. This is a well-known finding in the literature and should be put into perspective (cf. Lenzi 2018). Mitigation

deterrence is a logical implication of IAMs that optimize for whole century costs. To address this shortcoming, Rogelj et al. (2021) propose a scenario design to detach CDR deployment from emissions reductions. Applying this logic could make results more policy-relevant by providing policymakers with different options for specifying Net-Zero targets. As this is not the focus of the study, one might simply discuss the conclusions. The results are still relevant, as they quantify deterrence under a set of assumptions.

The Reviewer has raised another crucial issue regarding the need to address the mitigation deterrence associated with high reliance on CDR. Several studies, including those by Hoglund et al. ¹, McLaren et al. ², and Rogelj et al. ³, have discussed a potential strategy where separate emission reduction targets are set apart from CDR deployment, as clearly stated by the Reviewer. Incorporating this argument into our manuscript further highlights the distinction between high CDR pathways and moderate/low CDR pathways. In the high CDR pathway, there is no clear differentiation between emission reduction and CDR deployment. Following a least-cost approach, CDR deployment tends to grow endogenously without any predefined limit in order to achieve the climate target. However, in the low/moderate CDR pathways, our modeling approach involves setting distinct CDR targets based on available CDR options and maximum removal capacity.

Below, we have included relevant excerpts from the manuscript that elaborate on these points.

- (1) In the scenario where the reliance on CDR is modeled as HIGH, the complete suite of six CDR approaches is deployed from the beginning of policy implementation to the end of 2050. These CDR approaches include AR, Biochar, BECCS, DACCS, DORCS, and ERW, and they are deployed without constraining how much CO₂ they can remove from the atmosphere per year. With no limit on negative emissions, gross CDR deployment in Asia endogenously increases to about 12 GtCO₂/yr by 2050 in this scenario.

The obstruction of mitigation efforts when relying heavily on CDR is by design integrated into climate modeling⁴. CDR unavoidably displaces some of the immediate mitigation actions. Since scenarios in IAMs like GCAM are designed to minimize mitigation costs over the century, the inclusion of CDR alters the distribution of these costs throughout that timeframe. The introduction of CDR into IAMs results in higher emissions in the near term compared to scenarios without CDR^{5,6}. This increase implies that immediate climate action becomes less stringent and, consequently, less costly. This is not because CDR makes near-term mitigation more expensive, but rather because the availability of CDR reduces the overall cost of mitigation over the entire century^{4,7,8}.

Rogelj et al. proposed a framework that provides a logic to separate emission reduction from CDR deployment³. This framework enables studies to explore future CDR deployment as an independent variation under a desired temperature outcome. In that case, in an environment where emission reduction policies are independent of negative emissions, one cannot replace the other. Negative emissions would be available only for the specified amount required without delaying emission reduction^{1,2}. Based on these principles, we have developed two additional scenarios, namely, MODERATE and LOW, each with distinct CDR targets in terms of available CDR options and removal capacity. The descriptions of these two scenarios are provided below.

- (2) In the scenario where the reliance on CDR is modeled as MODERATE, only AR and BECCS are deployed. AR is endogenously deployed, and the rationale for choosing BECCS as the sole novel CDR over other alternatives is because BECCS is the most represented novel CDR in IAM-based studies and has relatively low cost and high maturity. BECCS can also provide low-carbon energy in the electricity, refinery, and hydrogen sectors as it captures CO₂. In this scenario, CDR has been deployed with a specified upper limit on how much CO₂ they can remove per year. The deployment of BECCS and the upper limit on gross negative emissions begin in 2025 and continues until the end of 2050. Here, global

BECCS deployment is exogenously limited to 2 GtCO₂/yr from 2025 to 2050. 2 GtCO₂/yr removal here is to represent the current gross CDR available globally⁹. Asia's share in this BECCS amount is endogenously deployed, reaching an annual average of 1.8 GtCO₂/yr from 2025 to 2050. In this scenario, total gross CO₂ removal in Asia reaches 2.3 GtCO₂/yr by 2050.

- (3) In the LOW scenario, only AR is deployed for negative emissions. Here, we limit CDR by novel means to 0 GtCO₂ throughout the modeling period. AR is endogenously deployed, where total gross CO₂ removal reaches 0.5 GtCO₂/yr by 2050.

We also emphasize the point further in under “policy implications”.

In this study, we highlight the risks of an overreliance on CDR leading to delayed decarbonization efforts in the near term. It is imperative for governments and other key stakeholders to prioritize policies that emphasize non-CDR mitigation strategies, while viewing carbon removal as a complementary tool. Policies should be structured in a manner that prevents decarbonization and carbon removal from substituting one another. This can be achieved through policy designs that separate CDR deployment from emission reduction objectives^{2,3}. Moreover, while decarbonization efforts should take precedence in achieving the 1.5°C target, CDR plays a crucial role in ensuring the success of end-of-century targets. The cost associated with CDR poses a significant limitation, and policies should be implemented to facilitate the rapid scaling of novel CDR technologies. Furthermore, it is essential for policies that establish carbon removal targets to be grounded in realism. These technologies are still in their infancy, with numerous uncertainties surrounding their effectiveness and scalability. Placing excessive reliance on the potential success of large-scale carbon removal carries significant risks and could prove detrimental to the climate system if these technologies fail to meet expectations¹⁰. We discuss these policies in detail as follows.

CDR moral hazard or mitigation deterrence has been widely discussed¹¹⁻¹⁴, and one of the key ways to pursue CDR without shifting attention from near-term deep decarbonization is to set separate CDR and emission reduction targets. Separate CDR and emission reduction targets could be a valuable strategy to prevent potential delays in decarbonization, as it would discourage relying on unproven future CDR deployment². Setting distinct CDR targets allows explicit focus on ramping up deployment and investment in negative emissions technologies and nature-based solutions, which facilitates scaling up CDR to levels required to achieve net-zero/net-negative emissions¹. In an environment where emission reduction policies are independent of negative emissions, one cannot be replaced with the other. Negative emissions would be available only for the specified amount required without delaying emission reduction^{1,2,15}. For example, the EU has proposed reducing net greenhouse gas emissions by 90% by 2040 compared to 1990 levels. This means the EU's fossil fuel and industrial sectors must emit less than 850 MtCO₂e/yr by 2040, while land-based and industrial carbon removal should reach 400 MtCO₂/yr¹⁶. With separate targets, the role of CDR shifts from primarily offsetting residual emissions to more of a complementary mitigation measure¹⁷. In other words, CDR deployment is not contingent on high fossil fuel use and emissions. Distinct targets enable greater transparency in tracking progress on emission reductions versus negative emissions². This allows for course correction if efforts lag in either domain.

Our findings highlight the importance of making room for the development and deployment of novel forms of negative emissions. Relying solely on conventional CDR in the form of afforestation and reforestation, could result in higher residual emissions, particularly from the industrial sector. Additionally, afforestation and reforestation projects are susceptible to pests, natural hazards, and extreme weather events^{18,19}, which limits their effectiveness as long-term CDR solutions. In conjunction with afforestation and reforestation, the adoption of novel CDR technologies like BECCS and DACCS could help offset these residual emissions. However, it is essential to note that

these novel CDR approaches are associated with high costs, which may hinder their development and deployment trajectory. Governments can play a pivotal role in incentivizing novel CDR deployment, as exemplified by the Inflation Reduction Act (IRA) of 2022. This legislation expanded the 45Q tax credit for direct air carbon capture by 360% (from \$50 to \$180/ton)²⁰.

While CDR technologies are required for achieving the 1.5°C target, it is crucial to establish realistic targets for carbon removal. Conservative estimates suggest that novel CDR technologies must reach approximately 4.2 (with a range of 3.7–6.2) gigatonnes of CO₂ per year by 2050 to remain on track for the 1.5°C target²¹. However, current deployment of these novel CDR technologies stands at only 0.002 gigatonnes of CO₂ per year⁹. This means that these technologies would need to increase by about 1800-fold between now and around 2050. To put this into perspective, the share of renewables in total energy supply from now to 2050 to achieve net zero emissions would need to increase by 5-fold, while final energy demand from hydrogen would need to increase by 3-fold between 2030 and 2050²². Relying on uncertain double-digit (>10 gigatonne) scale CDR between mid-century and the end of the century may lead to high temperature overshoot, with potential irreversible climate impacts¹⁰ if the expected >10 gigatonne CDR fails to materialize. Therefore, it is crucial for policymakers to approach carbon removal targets with caution and ensure that they are based on realistic assessments of technological capabilities and their sustainability issues. Given our relatively greater experience with decarbonization measures, policies should primarily focus on initiatives such as renewables, electrification, energy efficiency, and carbon-neutral fuels to account for 80-90% of the necessary reduction in net emissions. CDR will then play a crucial role in offsetting the remaining 10-20% of residual emissions, particularly from hard-to-abate sectors.

Revisions to the main text in response to the Reviewer’s suggestions can be found in discussions under “policy implications” and “scenario formulation”

(3) The paper would profit from reporting the discount rate and making transparent, in how far key findings are dependent on the chosen SDR. In the context of modeling CDR, cost-minimization and high discount rates are known to have a large impact on key results, such as mitigation costs, net-zero timing and overall deployment of negative emissions (cf. Emmerling et al. 2019; Grant et al. 2021; Riahi et al. 2021). The discount rate is not discussed in the current manuscript, though the authors refer to Fuhrmann et al. 2021 for discussions. Transparency of the results could be improved by including a short discussion. Varying the discount rate could alter central results such as the high demand for cropland and the significantly higher costs in Moderate and Low. However, this issue is not particular for this study but reflects a lack of transparency of integrated modeling overall.

The Reviewer accurately highlights the critical importance of the discount rate in mitigation scenarios generated by IAMs. The relevance of the discount rate has indeed been extensively discussed in previous studies²³⁻²⁵. We acknowledge that our manuscript could benefit from addressing this aspect, but it's essential to note that, based on our specific scenario settings – which involve imposing constraints on net emissions – our mitigation pathways are not influenced by the discount rate.

However, we agree that the choice of discount rate can have large influence in other aspects of the research, and as such we now added a discussion of the relevance of the discount rate and mention the default rate used in the GCAM model for the sake of transparency. It's important to emphasize that had our scenarios been structured differently to allow endogenous delays in mitigation, certain aspects related to the impact of the discount rate could have been explored. For instance, if our scenarios had specified a fixed end-of-century climate target, similar to the approach taken in the work of Fuhrman et al.²⁵, we could have investigated how the discount rate affects the outcomes. Conversely, if our

climate target had been formulated using a carbon pricing approach, we could have exogenously specified the discount rate. This is by design. We're not trying to assess the intertemporally "optimal" decision maker. We're simulating an imperfect decision making.. Below, we have included excerpts from the manuscript that shed light on these considerations.

The discount rate is an important factor to consider in climate mitigation scenarios, particularly those generated by IAMs. In climate change and IAMs, the discount rate refers to the rate at which future costs and benefits are discounted or given less weight compared to present costs and benefits²⁶. Most IAMs commonly employ a discount rate of 3% to 5%. However, there are alternative rates within this range and beyond. Previous studies, such as those conducted by Riahi et al.²⁴, Fuhrman et al.²⁵ and Emmerling et al.²³, demonstrated that lower discount rates tend to favor earlier mitigation actions, reducing the reliance on CDR technologies and temperature overshoot. Essentially, a lower discount rate emphasizes taking immediate steps to mitigate climate change, resulting in higher initial transition costs. Nevertheless, this approach yields long-term economic benefits and earlier advantages in avoiding climate change impacts. Conversely, higher discount rates tend to postpone mitigation efforts, leading to delayed transition pathways.

In GCAM, the standard discount rate used is either 3% or 5% (as used in this study). The discount rate represents the exogenous escalation in the price of CO₂ or GHG over time, either within a scenario with an end-of-century target or one with an exogenous carbon pricing trajectory²⁷. Investigating the role of discount rate in our mitigation pathways is beyond the scope of this study. Our objective is not to evaluate the intertemporally "optimal" decision-maker, but rather to simulate imperfect decision-making. In our case, where constraints are set on GHG emissions (as opposed to scenarios with fixed end-of-century climate target^{25,28,29}), the model simply needs to align with the pre-determined emissions constraint for each period regardless of the chosen discount

rate. That is, we do not allow the flexibility of optimally delaying mitigation based on high discount rate.

Revisions to the main text in response to the Reviewer's suggestions can be found in discussion under "sensitivity analysis".

(4) Readability of Fig. 7 might be improved by using absolute numbers for a-c given the low starting values.

We thank the Reviewer for their concerns. However, to maintain consistency among the sub-figures in Fig. 7 (now Fig. 6 in the revised manuscript), we prefer to retain the percentage shares instead of using absolute values. Nevertheless, we have made significant improvements to all the figures to enhance clarity and presentation.

We are truly grateful to the Reviewer for the time taken to review and carefully comment on our manuscript. Suggestions and recommendations have significantly improved the paper. We hope that our responses meet the Reviewer's expectations.

Reviewer 2

Carbon dioxide removal (CDR) is essential to achieve the net-zero emissions, and the deployment level of CDR technology is greatly depend on the time of net-zero emissions, all greenhouse gas emissions mitigation technologies, social and economic development, related resource constraints and etc. This paper focuses on the impacts of different CDR deployment level using GCAM model, and provides more perspective to consider CDR development. Although the topic is interesting, I think there are some issues with this paper, the authors are requested to go through the following comments.

The authors thank the Reviewer for their careful attention to our paper and the constructive feedback. We greatly appreciate you highlighting areas where we can strengthen our study design and writing. We have addressed each of your concerns in detail, and we hope that the revisions we have made meet your expectations. Please let us know if you have any other suggestions based on the revised version.

(1) My biggest concern is that the conclusion of this article is very obvious. By 2050, net-zero greenhouse gas emissions will be achieved. Under HIGH scenarios, greenhouse gas emissions will inevitably be higher, and fossil energy utilization will also be more. This is closely related to the scenario setting, thus the results in Table 2 are very obvious. The relationship between CDR and fossil energy consumption is consistent with other literature, and there are no new findings. Furthermore, the subsequent research results on land and water are also evident.

Overall, the analysis of this paper is not at a high level, only emphasizing the adverse impacts of CDR technology.

The authors acknowledge the Reviewer's concerns and we hope the clarifications made here and the revised manuscript would shed more light on the importance, timeliness and contribution of our study in moving the field forward.

(1) We would like to provide the key contributions we have made, followed by an explanation as to why our conclusions are not obvious.

First, with the existing studies on mitigation deterrence associated with CDR being qualitative and theoretical in nature ^{4,30-34}, we quantify the negative impacts on regional energy-land-water-food system associated with the expectation of tens of gigatonnes CDR deployment by mid-century.

Second, with Asia poised to become the region with the highest future carbon removal capacity³⁵, there has been no comprehensive study quantifying how individual Asian countries and regions would respond to the risks associated with mitigation deterrence linked to CDR deployment.

Also, an open question arises: Does a lower CDR scenario always yield the best outcome compared to a higher CDR scenario? What happens when a lower CDR scenario lacks room for implementing novel CDR approaches, while a relatively higher CDR scenario incorporates novel CDR approaches? In this paper, we address this question by refraining from deploying any novel CDR in the lowest CDR scenario. The results indicate that while a low CDR scenario is preferable to excessive CDR, it is not necessarily superior to a moderate CDR scenario. In other words, while an excessive amount of CDR is unnecessary, it does not imply that novel CDR approaches should be disregarded. A balanced approach to CDR is deemed optimal. This conclusion represents a key aspect of our study's contribution.

Furthermore, with increasing pressure to electrify and decarbonize our energy sector, many countries have installed large-scale electricity generation systems, with a majority relying on coal, natural gas, or oil. However, to meet climate targets, it is imperative to phase out these fossil-based electricity generation systems pre-maturely. This poses a significant challenge, as many of these systems have lifetimes ranging from 45 to 60 years. Consequently, countries that recently invested in these technologies face substantial financial losses as these assets become stranded. Existing studies on the issue of technology stranding have primarily focused on the role of decarbonization or emission reduction³⁶⁻⁴⁰. In this paper, we also examine how different levels of CDR can either alleviate or exacerbate the problem of stranded assets, particularly in regions such as Asia where there is significant carbon lock-in.

Additionally, we also show the role of land use sector's involvement in climate mitigation under different CDR deployment scenarios and how each level of involvement affects key sustainability indicators in Asia's pathway to net-zero emissions. Finally, with biochar gaining traction as a

potential CDR method with a mitigation potential exceeding 5 GtCO₂/yr²⁶, we also examine how biochar application and sequestration rates may affect regional sustainability.

(2) Also, the issue of mitigation deterrence (higher emissions and fossil fuel use) under CDR reliance is highly contested in the literature and not necessarily straightforward. The following arguments are quoted directly from the cited references.

- “While some have cautioned that a focus on CDR in policy processes may lead to delayed efforts to mitigate emissions, others have argued that such concerns are unwarranted. Nevertheless, the circumstances under which CDR could help or delay emissions mitigation in given contexts remain unclear” <https://doi.org/10.1016/j.envsci.2023.103659>
- “There is considerable disagreement on how serious a risk moral hazard or mitigation deterrence (henceforth “MH/MD”) actually is. Some maintain that CDR is already distracting from emission reductions, or point to how it might do so. Others find no evidence of moral hazard, suggest that the real-world risk might be overstated, or argue that suggestions of a potential trade-off between CDR and emission reductions are ultimately counterproductive” <https://doi.org/10.1002/wcc.826>

Most of the existing studies on this topic have been primarily theoretical, philosophical, and qualitative in nature^{4,30–34}. Also, to the best of our knowledge, no other study quantifies the negative impacts of excessive CDR reliance on energy-land-water system as opposed to pursuing deep decarbonization. Our contribution to the discussion thus lies in quantifying these impacts and providing a quantitative evidence to emphasize the importance of prioritizing deep decarbonization as the primary strategy, with CDR as a complementary measure.

It's essential to clarify that the mere HIGHER deployment of CDR DOES NOT lead to higher residual GHG emissions – excessive reliance rather DOES lead to higher residual GHG emissions. Our results, as presented in Supplementary Table S2 (Previously Table 3) confirms this. We demonstrate that having a gross CDR level of 2.3 GtCO₂ by 2050 (Moderate CDR pathway) can result in lower total residual GHG emissions compared to a scenario with gross CDR at 0.5 GtCO₂ (Low CDR pathway). This is because, without a sufficient amount of negative emissions,

residual emissions from hard-to-abate sectors, particularly the industrial sector, cannot be effectively offset, leading to higher residual GHG emissions. In essence, while an excessive reliance on CDR is not desirable, our findings highlight the necessity of incorporating some level of CDR to complement decarbonization measures in order to achieve a significant reduction in GHG emissions.

Below, we have included excerpts from the manuscript that elaborate on this particular point.

As shown in Fig. 3a, total positive emissions in 2050 are lower in the MODERATE (11 GtCO₂e/yr) and LOW (13 GtCO₂e/yr) scenarios than in the HIGH scenario (16 GtCO₂e/yr). Notably, in the absence of any novel means of CDR to offset residual emissions from the industrial sector, its decarbonization becomes increasingly difficult, leading to higher residual CO₂ emissions from the sector, as shown in Supplementary Table 2.

The comparison between the HIGH and LOW scenarios highlights that while prioritizing decarbonization is paramount, a 1.5°C climate pathway without adequate negative emissions to offset residual emissions, especially in the hard-to-abate industrial sector, as represented by the LOW scenario, could lead to higher residual GHG emissions than in the MODERATE scenario. We also find that while the LOW scenario realizes greater energy efficiency improvements compared to the HIGH scenario, the lack of adequate negative emissions under the former leads to minimal differences in energy efficiency gains between the LOW and MODERATE scenarios. These contrasting outcomes between the HIGH vs LOW scenarios and MODERATE vs LOW scenarios reinforce that the world must prioritize decarbonization but some negative emissions, especially from novel CDR, will likely be required to play a complementary role to decarbonization to provide a balanced and net positive impact. Our result indicates that pursuing CDR in the context of a deep decarbonized energy system, as represented by the MODERATE scenario, could reduce some of the most adverse impacts of negative emissions compared to pursuing CDR under a less

decarbonized energy system (HIGH) or pursuing deep decarbonization without allowing room for adequate CDR (LOW).

Revisions to the main text relating to the above discussions can be found in discussion under “Impact on positive and negative emissions, net zero timing, pollutants”.

(2) The application of any technology will have some impacts. I believe that for achieving the specific temperature control goal of 1.5 °C, which can avoid climate loss, CDR is one of the optimal technology combinations under many constraints such as technological availability and economy factors. Therefore, this article should explain and discuss what the constraints of CDR technology are considered in the model. For example, in the high scenario of 12GtCO₂, whether this setting takes into account resource potential and other related constraints, and whether it can be achieved..

The Reviewer's observation regarding the constraints on resources under CDR deployment is indeed accurate. Our primary objective in this study was to quantify the repercussions associated with varying levels of CDR reliance on regional energy-land-water system. Consequently, we did not explicitly impose constraints on resources. However, it's important to highlight that placing limits on CDR deployment indirectly restricts their potential negative impacts on natural resources (which is our ultimate goal of this study). For instance, limiting BECCS reduces competition for land and leads to the expansion of cropland (as demonstrated in Supplementary Table S4, Figure 5, and Figure 6). Similarly, limiting or avoiding DACCS reduces energy consumption associated with CDR (as shown in Supplementary Figure 1). Increasing the involvement of the land use sector in climate mitigation efforts results in the expansion of cropland (as depicted in Fig. 8a), among other outcomes.

Directly imposing constraints on resources would have been inconsistent with our study design, as we aimed to investigate how varying levels of CDR reliance might positively or

negatively affect these resources. Nevertheless, it's worth noting that GCAM provides default constraints on resources that users can modify for their investigations. For instance, users can set aside a portion of natural land, preventing it from being converted to crops or other land types. The default setting in our pathways was a 90% protection of all non-commercial ecosystems. Water constraints can also be imposed, which can exacerbate regional water scarcity. Additionally, constraints on fertilizer demand can be introduced⁴¹. However, these assessments were beyond the scope of our study, as they would have deviated from our primary focus on CDR and its impacts on Earth's system.

Nonetheless, the results presented in Supplementary Table 5 concerning biomass consumption, cropland allocation, and forest cover suggest that the resource-related impacts we observed were within the range of those seen in existing scenarios in the literature.

(3) The introduction of the model is very unclear, and it is unclear how the author improved the GCAM model to obtain the research results in the paper. There is a lack of explanation for the relevant models of land, water, and fertilizer.

We updated the model to achieve our research objectives in the following regard.

Several IAMs primarily focus on afforestation/reforestation (AR) and bioenergy with carbon capture and storage (BECCS) for modeling negative emissions. Only a few IAMs can model direct air carbon capture and storage (DACCS). AR, BECCS, and DACCS are the only three CDR approaches in the public release version of the Global Change Assessment Model (GCAM). To achieve our objectives, we modified the core version of GCAM5.4 to create our version called "GCAM-TJU." This modified version includes three additional novel CDR methods: biochar, enhanced rock weathering (ERW), and direct ocean carbon removal and storage (DORCS). Detailed descriptions and assumptions for all six CDR approaches are provided in the

Supplementary Discussion. (Section S5 and S6; Tables S7, S8, S9, S13, S14, S15, S16, S17, S18, S19, S20, S21, S22, S23, S24, S25 Fig. S28, Fig. S29) .

Previously, we had simply referenced the online documentation of GCAM. However, we agree that the key sectors relevant to this research should be described in detail in the paper. In the revised manuscript, we have included a more detailed description of the model in the Supplementary Material; (Section S6) and its land (Subsection S6.2.3), water (Subsection S6.2.4), fertilizer (Subsection S6.2.5) modules.

(4) In the high scenario, there are six CDR technologies required, some of CDR technology are expensive, then why is the carbon price \$248/tCO₂ in Table 2, which is the lowest among HIGH, MODERATE, and LOW scenarios. I have some doubts about this carbon price.

We acknowledge the Reviewer's concerns and we would like to clarify how the high CDR pathway is the least-cost pathway of the three. Yes, CDR technologies are expensive but the most expensive ones such as DORCS are not really being used at scale, rather the cheaper ones are the ones dominating the market. Furthermore, CDR such as DACCS can act as a backstop technology for climate change, meaning that it can provide a last-resort option to reduce CO₂ emissions when other mitigation options in hard-to-abate sectors are too expensive or impractical^{7,25}. Without CDR, the marginal abatement cost curve increases exponentially, as we approach 100% mitigation⁸. CDR availability has a larger impact on marginal cost savings, measured in terms of the CO₂ allowance price, than on total costs, and the flattening of the abatement cost curve with CDR leads to a linear increase of total costs in abatement effort (rather than a nonlinear increase without CDR)⁸. This is because hard-to-abate sectors like aviation and agriculture can use CDR technologies to offset some emissions and avoid the need for more expensive abatement options.

Without CDR in Asia's climate mitigation strategy, capital-intensive net zero GHG emitting technologies with increasingly low utilization rates compared to their fossil fuel counterparts,

would have to increase to reduce emissions. This ultimately increases marginal cost - making the HIGH CDR pathway the least expensive pathway.

-Lenzi et al.⁴ also provide the following explanation: “*NETs inevitably displace some near-term mitigation. Since scenarios aim to minimize mitigation costs over the century, the inclusion of NETs (an assumption labelled ‘full technological availability’ among modellers) alters the distribution of mitigation costs over the century. Introducing NETs into IAMs increases near-term emissions compared with a non-NETs scenario. This increase means that near-term climate action is less stringent and hence less costly. This is not because NETs make near-term mitigation more expensive, but because the availability of NETs lowers the aggregated cost of mitigation over the course of the century*”.

-Low carbon price under CDR-based pathways is widely reported in several studies⁵⁻⁸.

For clarity, we provided an explanation for the high CO₂ price in the LOW CDR pathways in the revised manuscript in discussion under “Impact on primary and final energy demand, and abatement costs” as follows:

Under limited availability of carbon removal, the marginal abatement cost of carbon (Supplementary Fig. 1b and Supplementary Table 1) is considerably higher in the MODERATE and LOW scenarios than in the HIGH scenario. CDR technologies, while expensive, have the potential to delay the urgent need for emission reduction⁴². In particular, DACCS can serve as a backstop technology for climate change mitigation, providing a last-resort option to reduce CO₂ emissions when other mitigation options in hard-to-abate sectors become prohibitively expensive or impractical⁷. Without CDR, the marginal abatement cost curve tends to increase exponentially as we approach 100% mitigation⁸, indicating that costs escalate rapidly as emissions reduction targets become more stringent. This arises from greater reliance on expensive technologies to eliminate the remaining emissions. Without CDR in Asia's climate strategy, capital-intensive net

zero GHG emitting technologies with increasingly low utilization rates compared to their fossil fuel counterparts would need to expand substantially to enable significant emission cuts⁶. This ultimately increases the marginal costs, making the HIGH scenario the least expensive pathway explored here. Low carbon price under CDR-based pathways is widely reported in several studies⁵⁻⁸.

(5) The scenario settings in the model lack explanation, such as reaching 12GtCO₂ by 2050 under HIGH scenario; In the MODERATE scenario, the annual average is 1.8 GtCO₂, while in the LOW scenario it is 0.5 GtCO₂ by 2050. The author needs further explanation on why this is set.

The reviewer rightfully points out that the previous version of the manuscript lacked clarity regarding the methodology used to derive these values. In the revised manuscript, we have addressed this issue with the following clarification:

- (1) In the scenario where the reliance on CDR is modeled as HIGH, the complete suite of six CDR approaches is deployed from the beginning of policy implementation to the end of 2050. These CDR approaches include AR, Biochar, BECCS, DACCS, DORCS, and ERW, and they are deployed without constraining how much CO₂ they can remove from the atmosphere per year. With no limit on negative emissions, gross CDR deployment in Asia ENDOGENOUSLY increases to 12 GtCO₂ by 2050 in this scenario.
- (2) In the scenario where the reliance on CDR is modeled as MODERATE, only AR and BECCS are deployed. AR is endogenously deployed, and the rationale for choosing BECCS as the sole novel CDR over other alternatives is because BECCS is the most represented novel CDR in IAM-based studies and has relatively low cost and high maturity. BECCS can also provide low-carbon energy in the electricity, refinery, and hydrogen sectors as it captures CO₂. In this scenario, CDR has been deployed with a specified upper limit on how

much CO₂ they can remove per year. The deployment of BECCS and the upper limit on gross negative emissions begin in 2025 and continues until the end of 2050. Here, global BECCS deployment is EXOGENOUSLY limited to 2 GtCO₂/yr from 2025 to 2050. 2 GtCO₂/yr removal here is to represent the current gross CDR available globally⁹. Asia's share in this BECCS amount is ENDOGENOUSLY deployed, reaching an annual average of 1.8 GtCO₂ from 2025 to 2050. In this scenario, total gross CO₂ removal in Asia reaches 2.3 GtCO₂ by 2050.

- (3) In the LOW CDR scenario, only AR is deployed for negative emissions. Here, we limit CDR by novel means to 0 GtCO₂ throughout the modeling period. AR is ENDOGENOUSLY deployed, where total gross CO₂ removal reaches 0.5 GtCO₂ by 2050.

Revisions to the main text in response to the Reviewer's suggestions can be found in discussion under "scenario formulation".

(6) The figures in the paper are not clear.

We have made significant improvements to all the figures to enhance clarity and presentation.

(7) The paper lacks some references, such as line64-72.

We value the Reviewer's feedback. In the revised version, we have provided references in the lines suggested by the Reviewer and also double-checked that all sources of information in our manuscript are properly cited.

(8) There are multiple statements in the paper regarding over reliance on CDR. How is the definition of over reliance defined here, whether it is the proportion of emissions reduction exceeding or how it is defined.

We appreciate the Reviewer's feedback. In the context of our study, the term "CDR over-reliance" does not indicate a fixed proportion of emission reductions or represent a specific threshold value beyond which CDR reliance becomes excessive. Instead, in our study, "over-reliance on CDR" refers to any level of dependence on large-scale CDR towards tens of gigatonne removal by mid-century. At this level of reliance, the deployment of CDR could have adverse effects on global sustainability. This excessive reliance on CDR might result in the continuation of fossil fuel emissions, mitigation of immediate climate actions, and potential negative impacts on the climate-energy-land-food-water system.

In the context of our study specifically, we define CDR reliance as "over-reliance" when the deployment of these strategies reaches at least 10 GtCO₂/yr, particularly in the first half of this century.

(9) Policy implications are not in-depth enough. For example, non-CDR strategies as renewable ... are too general, authors should clarify specific measures.

We appreciate the Reviewer's feedback and suggestions. Incorporating specific measures to address the issues raised in our paper will certainly enhance the contribution of our work to the existing literature. In the discussions under "policy implications", we have emphasized the policy implications and outlined specific measures from three distinct angles:

1. Separate CDR and emission reduction targets.
2. Policies to cut CDR cost
3. Realistic CDR targets.

In this study, we highlight the risks of an overreliance on CDR leading to delayed decarbonization efforts in the near term. It is imperative for governments and other key stakeholders to prioritize policies that emphasize non-CDR mitigation strategies, while viewing carbon removal

as a complementary tool. Policies should be structured in a manner that prevents decarbonization and carbon removal from substituting one another. This can be achieved through policy designs that separate CDR deployment from emission reduction objectives ^{2,3}. Moreover, while decarbonization efforts should take precedence in achieving the 1.5°C target, CDR plays a crucial role in ensuring the success of end-of-century targets. The cost associated with CDR poses a significant limitation, and policies should be implemented to facilitate the rapid scaling of novel CDR technologies. Furthermore, it is essential for policies that establish carbon removal targets to be grounded in realism. These technologies are still in their infancy, with numerous uncertainties surrounding their effectiveness and scalability. Placing excessive reliance on the potential success of large-scale carbon removal carries significant risks and could prove detrimental to the climate system if these technologies fail to meet expectations ¹⁰. We discuss these policies in detail as follows.

CDR moral hazard or mitigation deterrence has been widely discussed ¹¹⁻¹⁴, and one of the key ways to pursue CDR without shifting attention from near-term deep decarbonization is to set separate CDR and emission reduction targets. Separate CDR and emission reduction targets could be a valuable strategy to prevent potential delays in decarbonization, as it would discourage relying on unproven future CDR deployment ². Setting distinct CDR targets allows explicit focus on ramping up deployment and investment in negative emissions technologies and nature-based solutions, which facilitates scaling up CDR to levels required to achieve net-zero/net-negative emissions ¹. In an environment where emission reduction policies are independent of negative emissions, one cannot be replaced with the other. Negative emissions would be available only for the specified amount required without delaying emission reduction ^{1,2,15}. For example, the EU has proposed reducing net greenhouse gas emissions by 90% by 2040 compared to 1990 levels. This means the EU's fossil fuel and industrial sectors must emit less than 850 MtCO₂e/yr by 2040, while land-based and industrial carbon removal should reach 400 MtCO₂/yr ¹⁶. With separate targets, the

role of CDR shifts from primarily offsetting residual emissions to more of a complementary mitigation measure ¹⁷. In other words, CDR deployment is not contingent on high fossil fuel use and emissions. Distinct targets enable greater transparency in tracking progress on emission reductions versus negative emissions ². This allows for course correction if efforts lag in either domain.

Our findings highlight the importance of making room for the development and deployment of novel forms of negative emissions. Relying solely on conventional CDR in the form of afforestation and reforestation, could result in higher residual emissions, particularly from the industrial sector. Additionally, afforestation and reforestation projects are susceptible to pests, natural hazards, and extreme weather events ^{18,19}, which limits their effectiveness as long-term CDR solutions. In conjunction with afforestation and reforestation, the adoption of novel CDR technologies like BECCS and DACCS could help offset these residual emissions. However, it is essential to note that these novel CDR approaches are associated with high costs, which may hinder their development and deployment trajectory. Governments can play a pivotal role in incentivizing novel CDR deployment, as exemplified by the Inflation Reduction Act (IRA) of 2022. This legislation expanded the 45Q tax credit for direct air carbon capture by 360% (from \$50 to \$180/ton) ²⁰.

While CDR technologies are required for achieving the 1.5°C target, it is crucial to establish realistic targets for carbon removal. Conservative estimates suggest that novel CDR technologies must reach approximately 4.2 (with a range of 3.7–6.2) gigatonnes of CO₂ per year by 2050 to remain on track for the 1.5°C target ²¹. However, current deployment of these novel CDR technologies stands at only 0.002 gigatonnes of CO₂ per year ⁹. This means that these technologies would need to increase by about 1800-fold between now and around 2050. To put this into perspective, the share of renewables in total energy supply from now to 2050 to achieve net zero emissions would need to increase by 5-fold, while final energy demand from hydrogen would need to increase by 3-fold between 2030 and 2050 ²². Relying on uncertain double-digit (>10 gigatonne)

scale CDR between mid-century and the end of the century may lead to high temperature overshoot, with potential irreversible climate impacts ¹⁰ if the expected >10 gigatonne CDR fails to materialize. Therefore, it is crucial for policymakers to approach carbon removal targets with caution and ensure that they are based on realistic assessments of technological capabilities and their sustainability issues. Given our relatively greater experience with decarbonization measures, policies should primarily focus on initiatives such as renewables, electrification, energy efficiency, and carbon-neutral fuels to account for 80-90% of the necessary reduction in net emissions. CDR will then play a crucial role in offsetting the remaining 10-20% of residual emissions, particularly from hard-to-abate sectors.

Revisions to the main text in response to the Reviewer's suggestions can be found in discussion under "policy implications".

We are truly grateful to the Reviewer for the time taken to review and carefully comment on our manuscript. Suggestions and recommendations have significantly improved the paper. We hope that our responses meet the Reviewer's expectations.

Reviewer 3

This paper models future decarbonization scenarios that rely heavily on NETs/CDR. The work is of contemporary interest and is scientifically novel, but this version of the manuscript has flaws which need to be fixed prior to publication..

The authors thank the Reviewer for their careful attention to our paper and the constructive feedback. We greatly appreciate you highlighting areas where we can strengthen our study design and writing. We have addressed each of your concerns in detail, and we hope that the revisions we have made meet your expectations. Please let us know if you have any other suggestions based on the revised version.

(1) Specific assumptions used for the NETs covered in the work are unclear. For example, it is unclear if environmental co-benefits are included in the calculations. These should be specified for full transparency.

We appreciate the Reviewer's concern regarding the modeling assumptions in our study. The assumptions used for modeling various CDR approaches are provided in the supplementary material and discussed in detail. We have added details on specific assumptions used for NETs covered in the work in SI (Section S5 and S6; Tables S7, S8, S9, S13, S14, S15, S16, S17, S18, S19, S20, S21, S22, S23, S24, S25 Fig. S28, Fig. S29) .

While the assumptions for co-benefits and co-damages are not directly integrated into the modeling of the CDR technologies in our study, certain key co-benefits and co-damages relevant to model dynamics are indirectly derived in the results. For instance, some of the co-benefits included in the analysis encompass soil health, air pollution, water scarcity, mitigation cost, and energy output. As an example, in our high CDR pathway, we observe higher levels of air pollutants, a lower mitigation cost, and fewer issues related to stranded assets in the power sector. These results provide insights into the co-benefits and co-damages associated with different levels of CDR deployment, even though they are not explicitly included in the CDR modeling assumptions.

We now discuss this in detail in discussions under “Impact on primary and final energy demand, and abatement costs”, “Impact on positive and negative emissions, net zero timing, pollutants”, and “Impact on land, water, and fertilizer consumption”.

(2) The results of the work should be compared with prior work, and any discrepancies should be explained.

We appreciate the Reviewer's suggestion to benchmark our study against prior research. Comparing our results with those from previous studies can offer valuable insights into the reliability and robustness of our findings. We acknowledge this suggestion, and we have added

such a comparison in our manuscript (discussion under “results benchmarking”). Furthermore, we have discussed the reasons for discrepancies in certain results between our study and existing research to provide a more comprehensive understanding of our findings in the context of the broader body of work in this field.

Supplementary Table 5 provides a comprehensive overview of how our results compare to existing scenarios sourced from the IPCC AR6 database ⁴³. The existing data corresponds to the R5ASIA dataset. The R5ASIA dataset encompasses approximately 550 scenarios derived from 58 distinct models and their respective versions. However, the data shown in Supplementary Table 5, consisting of roughly 40 scenarios originating from 15 distinct models and their versions, exclusively focuses on scenarios that explicitly outline pathways in alignment with the global 1.5°C target, maintaining consistency with our own pathways.

The discrepancies in some indicators between our results and those from existing scenarios can mainly be attributed to what Dekker et al. ⁴⁴ referred to as 'energy model fingerprints.' These fingerprints describe how models differ in structure, objectives, assumptions, parameterization, and level of detail. These differences result in variations in the computed energy and climate policy scenarios ^{44,45}. Another reason could be how we categorize countries/regions under 'Asia' in our pathways compared to existing scenarios. For instance, the R5ASIA scenario does not include Japan in its scope (See Supplementary Table 6 for a complete list of Asian countries/regions under GCAM-TJU and that of R5ASIA). Additionally, the variation in mitigation pathways between our scenarios and the existing ones can significantly affect key modeling results. To illustrate, our pathway is one in which Asia collectively achieves net-zero GHG emissions by 2050. In contrast, the existing scenarios mainly follow a net-zero CO₂ pathway that typically supports a global net-zero by 2050, in line with the 1.5°C target. Consequently, the level of stringency in our pathways is of a higher magnitude than in the existing scenarios.

Supplementary Table 5 Comparison of results here with existing mitigation pathways for Asia

Indicator	Existing literature	Current study		
		High CDR	Moderate CDR	Low CDR
2050 power capacity additions (GW/yr)				
Coal	38-1420	1728	1688	1742
Solar	600-4088	3130	4913	5045
Nuclear	203-1306	1150	2136	2241
Biomass	11-966	475	252	128
2050 Sequestration (GtCO ₂ /yr)				
BECCS	0.1-7.1	4.6	1.9	0
Fossil CCS	0-7.5	7.1	2.0	2.4
DACCS	0-1.6	4.2	0	0
Afforestation	0.3-2.6	0.3	0.5	0.5
2050 Residual emissions (GtCO ₂ /yr)				
Energy and industry	-4.9 to 10.5	11	7.0	8.4
2050 Energy (EJ/yr)				
Final energy	72-213	328	226	226
Primary energy	109-373	450	312	313
2050 Primary energy (EJ)				
Biomass	11-112	92	106	122
2050 Land (million ha)				
Cropland	91-699	130	410	400
Forest	316-642	550	540	550
2050 Policy cost (US\$2010/tCO ₂)				
Carbon price	417-255134	296	2355	3093
2050 Water (km ³)				
Total water consumption	604	1399	1344	1336

(3) In general, the figures are of poor quality and need to be improved for easier reading.

We have made significant improvements to all the figures to enhance clarity and presentation.

(4) A few minor (typographical/grammatical) errors in the text should be corrected.

The entire manuscript has undergone proofreading by the authors, and corrections have been implemented as needed to ensure that the language and writing meet acceptable standards for publication in Nature Communications.

We are truly grateful to the Reviewer for the time taken to review and carefully comment on our manuscript. Suggestions and recommendations have significantly improved the paper. We hope that our responses meet the Reviewer's expectations.

References

- (1) Höglund, R.; Mitchell-Larson, E.; Delerce, S. *How to Scale Carbon Removal without Undermining Emission Cuts*; Carbon Gap, 2023. <https://carbongap.org/how-to-scale-carbon-removal-without-undermining-emission-cuts/> (accessed 2023-10-04).
- (2) McLaren, D. P.; Tyfield, D. P.; Willis, R.; Szerszynski, B.; Markusson, N. O. Beyond “Net-Zero”: A Case for Separate Targets for Emissions Reduction and Negative Emissions. *Front. Clim.* **2019**, *1*.
- (3) Rogelj, J.; Huppmann, D.; Krey, V.; Riahi, K.; Clarke, L.; Gidden, M.; Nicholls, Z.; Meinshausen, M. A New Scenario Logic for the Paris Agreement Long-Term Temperature Goal. *Nature* **2019**, *573* (7774), 357–363. <https://doi.org/10.1038/s41586-019-1541-4>.
- (4) Lenzi, D. The Ethics of Negative Emissions. *Glob. Sustain.* **2018**, *1*, e7. <https://doi.org/10.1017/sus.2018.5>.
- (5) Strefler, J.; Bauer, N.; Humpenöder, F.; Klein, D.; Popp, A.; Kriegler, E. Carbon Dioxide Removal Technologies Are Not Born Equal. *Environ. Res. Lett.* **2021**, *16* (7), 074021. <https://doi.org/10.1088/1748-9326/ac0a11>.
- (6) Ampah, J. D.; Jin, C.; Afrane, S.; Li, B.; Adun, H.; Liu, H.; Yao, M.; Morrow, D. Does China’s Pathway to Carbon Neutrality Require the Integration of Land-Based Biological Negative Emission Solutions with Geochemical and Chemical Alternatives? *Sustain. Prod. Consum.* **2024**, *45*, 27–41. <https://doi.org/10.1016/j.spc.2023.12.025>.
- (7) Realmonte, G.; Drouet, L.; Gambhir, A.; Glynn, J.; Hawkes, A.; Köberle, A. C.; Tavoni, M. An Inter-Model Assessment of the Role of Direct Air Capture in Deep Mitigation Pathways. *Nat. Commun.* **2019**, *10* (1), 3277. <https://doi.org/10.1038/s41467-019-10842-5>.
- (8) Bistline, J. E. T.; Blanford, G. J. Impact of Carbon Dioxide Removal Technologies on Deep Decarbonization of the Electric Power Sector. *Nat. Commun.* **2021**, *12* (1), 3732. <https://doi.org/10.1038/s41467-021-23554-6>.
- (9) Smith, S. M.; Geden, O.; Nemet, G. F.; Gidden, M. J.; Lamb, W. F.; Powis, C.; Bellamy, R.; Callaghan, M. W.; Cowie, A.; Cox, E.; Fuss, S.; Gasser, T.; Grassi, G.; Greene, J.; Lück, S.;

- Mohan, A.; Müller-Hansen, F.; Peters, G. P.; Pratama, Y.; Repke, T.; Riahi, K.; Schenuit, F.; Steinhilber, J.; Strefler, J.; Valenzuela, J. M.; Minx, J. C. *The State of Carbon Dioxide Removal - 1st Edition*; The State of Carbon Dioxide Removal, 2023. <https://doi.org/10.17605/OSF.IO/W3B4Z>.
- (10) Anderson, K.; Buck, H. J.; Fuhr, L.; Geden, O.; Peters, G. P.; Tamme, E. Controversies of Carbon Dioxide Removal. *Nat. Rev. Earth Environ.* **2023**, 1–7. <https://doi.org/10.1038/s43017-023-00493-y>.
- (11) Hart, P. S.; Campbell-Arvai, V.; Wolske, K. S.; Raimi, K. T. Moral Hazard or Not? The Effects of Learning about Carbon Dioxide Removal on Perceptions of Climate Mitigation in the United States. *Energy Res. Soc. Sci.* **2022**, *89*, 102656. <https://doi.org/10.1016/j.erss.2022.102656>.
- (12) Merk, C.; Pönitzsch, G.; Rehdanz, K. Do Climate Engineering Experts Display Moral-Hazard Behaviour? *Clim. Policy* **2019**, *19* (2), 231–243. <https://doi.org/10.1080/14693062.2018.1494534>.
- (13) Jebari, J.; Táiwò, O. O.; Andrews, T. M.; Aquila, V.; Beckage, B.; Belaia, M.; Clifford, M.; Fuhrman, J.; Keller, D. P.; Mach, K. J.; Morrow, D. R.; Raimi, K. T.; Visioni, D.; Nicholson, S.; Trisos, C. H. From Moral Hazard to Risk-Response Feedback. *Clim. Risk Manag.* **2021**, *33*, 100324. <https://doi.org/10.1016/j.crm.2021.100324>.
- (14) Baumgartner, T. *CDReality: Is CDR a “Moral Hazard”?* OpenAir Collective. <https://openaircollective.cc/cdreality-is-cdr-a-moral-hazard/> (accessed 2023-09-24).
- (15) Lee, K.; Fyson, C.; Schleussner, C.-F. Fair Distributions of Carbon Dioxide Removal Obligations and Implications for Effective National Net-Zero Targets. *Environ. Res. Lett.* **2021**, *16* (9), 094001. <https://doi.org/10.1088/1748-9326/ac1970>.
- (16) European Commission. *Europe’s 2040 Climate Target and Path to Climate Neutrality by 2050 Building a Sustainable, Just and Prosperous Society*; COM(2024) 63 final; Communication from the Commission to the European Parliament, the Council, the European Economic and Social Committee and the Committee of the Regions, 2024.
- (17) Adun, H.; Ampah, J. D.; Bamisile, O.; Hu, Y. The Synergistic Role of Carbon Dioxide Removal and Emission Reductions in Achieving the Paris Agreement Goal. *Sustain. Prod. Consum.* **2024**. <https://doi.org/10.1016/j.spc.2024.01.004>.
- (18) Liu, Z.; Deng, Z.; He, G.; Wang, H.; Zhang, X.; Lin, J.; Qi, Y.; Liang, X. Challenges and Opportunities for Carbon Neutrality in China. *Nat. Rev. Earth Environ.* **2021**, *3* (2), 141–155. <https://doi.org/10.1038/s43017-021-00244-x>.
- (19) Ludden, C. *Fit for Purpose? Assessing the Potential of Current Governance Approaches to Carbon Dioxide Removal in China, the United States and the European Union*; doi: 10.18449/2022WP09; German Institute for International and Security Affairs (SWP), 2022. https://www.swp-berlin.org/publications/products/arbeitspapiere/Ludden_CDR_Governance_in_China__the_US_and_the_EU.pdf.
- (20) Bipartisan Policy. Inflation Reduction Act (IRA) Summary: Energy and Climate Provisions. *Wash. DC Verfügbar Unter Httpsbipartisanpolicy Orgdownload* **2022**.
- (21) UNEP. *Emissions Gap Report 2023*; 2023. <http://www.unep.org/resources/emissions-gap-report-2023> (accessed 2023-11-28).
- (22) IEA. *Net Zero by 2050 – Analysis*; 2021. <https://www.iea.org/reports/net-zero-by-2050> (accessed 2023-08-10).
- (23) Emmerling, J.; Drouet, L.; Wijst, K.-I. van der; Vuuren, D. van; Bosetti, V.; Tavoni, M. The Role of the Discount Rate for Emission Pathways and Negative Emissions. *Environ. Res. Lett.* **2019**, *14* (10), 104008. <https://doi.org/10.1088/1748-9326/ab3cc9>.

- (24) Riahi, K.; Bertram, C.; Huppmann, D.; Rogelj, J.; Bosetti, V.; Cabardos, A.-M.; Deppermann, A.; Drouet, L.; Frank, S.; Fricko, O.; Fujimori, S.; Harmsen, M.; Hasegawa, T.; Krey, V.; Luderer, G.; Paroussos, L.; Schaeffer, R.; Weitzel, M.; van der Zwaan, B.; Vrontisi, Z.; Longa, F. D.; Després, J.; Fosse, F.; Fragkiadakis, K.; Gusti, M.; Humpenöder, F.; Keramidas, K.; Kishimoto, P.; Kriegler, E.; Meinshausen, M.; Nogueira, L. P.; Oshiro, K.; Popp, A.; Rochedo, P. R. R.; Ünlü, G.; van Ruijven, B.; Takakura, J.; Tavoni, M.; van Vuuren, D.; Zakeri, B. Cost and Attainability of Meeting Stringent Climate Targets without Overshoot. *Nat. Clim. Change* **2021**, *11* (12), 1063–1069. <https://doi.org/10.1038/s41558-021-01215-2>.
- (25) Fuhrman, J.; Clarens, A.; Calvin, K.; Doney, S. C.; Edmonds, J. A.; O'Rourke, P.; Patel, P.; Pradhan, S.; Shobe, W.; McJeon, H. The Role of Direct Air Capture and Negative Emissions Technologies in the Shared Socioeconomic Pathways towards +1.5 °C and +2 °C Futures. *Environ. Res. Lett.* **2021**, *16* (11), 114012. <https://doi.org/10.1088/1748-9326/ac2db0>.
- (26) IPCC. *Climate Change 2022: Mitigation of Climate Change. Contribution of Working Group III to the Sixth Assessment Report of the Intergovernmental Panel on Climate Change*; [P.R. Shukla, J. Skea, R. Slade, A. Al Khourdajie, R. van Diemen, D. McCollum, M. Pathak, S. Some, P. Vyas, R. Fradera, M. Belkacemi, A. Hasija, G. Lisboa, S. Luz, J. Malley, (eds.)]. Cambridge University Press, Cambridge, UK and New York, NY, USA, 2022.
- (27) Bond-Lamberty, B.; Patel, P.; Lurz, J.; Smith, S.; Synder, A.; Kyle, P. JGCR/Gcam-Core: GCAM 5.4. **2021**. <https://doi.org/10.5281/zenodo.5093192>.
- (28) Liu, H.; Ampah, J. D.; Afrane, S.; Adun, H.; Jin, C.; Yao, M. Deployment of Hydrogen in Hard-to-Abate Transport Sectors under Limited Carbon Dioxide Removal (CDR): Implications on Global Energy-Land-Water System. *Renew. Sustain. Energy Rev.* **2023**, *184*, 113578. <https://doi.org/10.1016/j.rser.2023.113578>.
- (29) Pradhan, S.; Shobe, W. M.; Fuhrman, J.; McJeon, H.; Binsted, M.; Doney, S. C.; Clarens, A. F. Effects of Direct Air Capture Technology Availability on Stranded Assets and Committed Emissions in the Power Sector. *Front. Clim.* **2021**, *3*, 660787. <https://doi.org/10.3389/fclim.2021.660787>.
- (30) Preston, C. J. Ethics and Geoengineering: Reviewing the Moral Issues Raised by Solar Radiation Management and Carbon Dioxide Removal. In *The Ethics of Nanotechnology, Geoengineering, and Clean Energy*; Routledge, 2017.
- (31) Cooley, S. R.; Klinsky, S.; Morrow, D. R.; Satterfield, T. Sociotechnical Considerations About Ocean Carbon Dioxide Removal. *Annu. Rev. Mar. Sci.* **2023**, *15* (1), 41–66. <https://doi.org/10.1146/annurev-marine-032122-113850>.
- (32) Andrews, T. M.; Delton, A. W.; Kline, R. Anticipating Moral Hazard Undermines Climate Mitigation in an Experimental Geoengineering Game. *Ecol. Econ.* **2022**, *196*, 107421. <https://doi.org/10.1016/j.ecolecon.2022.107421>.
- (33) Fuss, S.; Canadell, J. G.; Peters, G. P.; Tavoni, M.; Andrew, R. M.; Ciais, P.; Jackson, R. B.; Jones, C. D.; Kraxner, F.; Nakicenovic, N.; Le Quéré, C.; Raupach, M. R.; Sharifi, A.; Smith, P.; Yamagata, Y. Betting on Negative Emissions. *Nat. Clim. Change* **2014**, *4* (10), 850–853. <https://doi.org/10.1038/nclimate2392>.
- (34) Morrow, D. R.; Thompson, M. S.; Anderson, A.; Batres, M.; Buck, H. J.; Dooley, K.; Geden, O.; Ghosh, A.; Low, S.; Njamnshi, A.; Noël, J.; Táiwò, O. O.; Talati, S.; Wilcox, J. Principles for Thinking about Carbon Dioxide Removal in Just Climate Policy. *One Earth* **2020**, *3* (2), 150–153. <https://doi.org/10.1016/j.oneear.2020.07.015>.
- (35) Fuhrman, J.; Bergero, C.; Weber, M.; Monteith, S.; Wang, F. M.; Clarens, A. F.; Doney, S. C.; Shobe, W.; McJeon, H. Diverse Carbon Dioxide Removal Approaches Could Reduce Impacts on the Energy–Water–Land System. *Nat. Clim. Change* **2023**. <https://doi.org/10.1038/s41558-023-01604-9>.

- (36) Binsted, M.; Iyer, G.; Edmonds, J.; Vogt-Schilb, A.; Arguello, R.; Cadena, A.; Delgado, R.; Feijoo, F.; Lucena, A. F. P.; McJeon, H.; Miralles-Wilhelm, F.; Sharma, A. Stranded Asset Implications of the Paris Agreement in Latin America and the Caribbean. *Environ. Res. Lett.* **2020**, *15* (4), 044026. <https://doi.org/10.1088/1748-9326/ab506d>.
- (37) Auger, T.; Trüby, J.; Balcombe, P.; Staffell, I. The Future of Coal Investment, Trade, and Stranded Assets. *Joule* **2021**, *5* (6), 1462–1484. <https://doi.org/10.1016/j.joule.2021.05.008>.
- (38) Fofrich, R.; Tong, D.; Calvin, K.; Boer, H. S. D.; Emmerling, J.; Fricko, O.; Fujimori, S.; Luderer, G.; Rogelj, J.; Davis, S. J. Early Retirement of Power Plants in Climate Mitigation Scenarios. *Environ. Res. Lett.* **2020**, *15* (9), 094064. <https://doi.org/10.1088/1748-9326/ab96d3>.
- (39) Cui, R. Y.; Hultman, N.; Edwards, M. R.; He, L.; Sen, A.; Surana, K.; McJeon, H.; Iyer, G.; Patel, P.; Yu, S.; Nace, T.; Shearer, C. Quantifying Operational Lifetimes for Coal Power Plants under the Paris Goals. *Nat. Commun.* **2019**, *10* (1), 4759. <https://doi.org/10.1038/s41467-019-12618-3>.
- (40) Iyer, G.; Ledna, C.; Clarke, L.; Edmonds, J.; McJeon, H.; Kyle, P.; Williams, J. H. Measuring Progress from Nationally Determined Contributions to Mid-Century Strategies. *Nat. Clim. Change* **2017**, *7* (12), 871–874. <https://doi.org/10.1038/s41558-017-0005-9>.
- (41) Sinha, E.; Calvin, K. V.; Kyle, P. G.; Hejazi, M. I.; Waldhoff, S. T.; Huang, M.; Vishwakarma, S.; Zhang, X. Implication of Imposing Fertilizer Limitations on Energy, Agriculture, and Land Systems. *J. Environ. Manage.* **2022**, *305*, 114391. <https://doi.org/10.1016/j.jenvman.2021.114391>.
- (42) Fuhrman, J.; Clarens, A. F.; McJeon, H.; Patel, P.; Ou, Y.; Doney, S. C.; Shobe, W. M.; Pradhan, S. The Role of Negative Emissions in Meeting China’s 2060 Carbon Neutrality Goal. *Oxf. Open Clim. Change* **2021**, *1* (1), kgab004. <https://doi.org/10.1093/oxfclm/kgab004>.
- (43) Byers, E.; Krey, V.; Kriegler, E.; Riahi, K.; Schaeffer, R. *AR6 Scenario Explorer and Database hosted by IIASA*. <https://data.ece.iiasa.ac.at/ar6/#/workspaces/2123> (accessed 2023-07-11).
- (44) Dekker, M. M.; Daioglou, V.; Pietzcker, R.; Rodrigues, R.; de Boer, H.-S.; Dalla Longa, F.; Drouet, L.; Emmerling, J.; Fattahi, A.; Fotiou, T.; Fragkos, P.; Fricko, O.; Gusheva, E.; Harmsen, M.; Huppmann, D.; Kannavou, M.; Krey, V.; Lombardi, F.; Luderer, G.; Pfenninger, S.; Tsiropoulos, I.; Zakeri, B.; van der Zwaan, B.; Usher, W.; van Vuuren, D. Identifying Energy Model Fingerprints in Mitigation Scenarios. *Nat. Energy* **2023**, *8* (12), 1395–1404. <https://doi.org/10.1038/s41560-023-01399-1>.
- (45) Ou, Y. Decoding Energy Model Variations. *Nat. Energy* **2023**, *8* (12), 1309–1310. <https://doi.org/10.1038/s41560-023-01402-9>.

REVIEWER COMMENTS

Reviewer #1 (Remarks to the Author):

Thank you for the detailed responses to my comments and for incorporating them into the revised manuscript. My concerns are sufficiently addressed in the revised manuscript. The added section explain mitigation deterrence and its role in this study carefully. Moreover, the influence of the SDR is put into perspective, both by your reply and the added text. The sensitivity analysis provides a significant contribution to the robustness of the results in regards to central parameters.

Reviewer #2 (Remarks to the Author):

According to the reviewer's comments, the author has made corresponding modifications to the paper. However, based on the current version, I do not think it is a high-level paper and I cannot accept its publication. Mainly in the following aspects:

(1) Is the title "Deployment expectations of multi gigatonnes scale carbon dioxide removal could adaptively impact the global climate system" appropriate? It seems that the paper did not answer such a question. Firstly, the paper quantifies the impact of different scales of CDR technology on the energy-land-water-food system to limit warming to achieve net zero greenhouse gas emissions by 2050. Secondly, there are many indicators about the global climate system, including the hydrosphere, atmosphere, etc. from a natural science perspective, but these related indicators are not reflected in the paper. If the author refers to the impact on low-carbon transformation, but the prerequisite for the paper is to achieve net zero greenhouse gas emissions by 2050, and the related result analysis only extends to 2050, and there is no impact on the period after 2050. Therefore, the relationship between the title and the research content is not very matching.

(2) The abstract is too general to show the innovation of the paper, and the research conclusions are obvious, lacking new findings.

(3) The logic of the paper is not clear, especially the stranded asset analysis in the power sector in the revised manuscript, which confuses me. In fact, if the author wants to analyze the impact of negative carbon technology on sustainable development from the perspective of negative carbon technology, it should establish a complete sustainable development framework, and discuss the social, economic, environmental, climate, and other impacts of negative carbon technology within this unified framework. However, the problem studied in the paper is not from the perspective of sustainable development, so what is the significance of adding this part of research? In addition, there is no method description for stranded asset in the Methods section. Stranded asset depends on the detailed information of thermal power plants. It is unknown how the author obtained parameters such as operating time of different power plants in different countries.

(3) Line82-83, "Here, we show that deploying tens of gigatonnes of negative emissions by the mid-century could seriously impede climate goals and threatens sustainability across Asia" How can

this sentence be derived and how to imbed climate goals? Moreover, sustainable development is a comprehensive concept. Can the analysis of only a few aspects in the paper support this statement?

(4) Suggest the author to clarify the reasons for the increase in NH₃ emissions in the low and medium scenarios of Line 181-182, as well as the relatively high scenarios?

(5) The impact of emission reduction paths on emission reduction technologies is significant, and the Methodology section of the paper only considers one path, namely "total GHG emissions in the region peak before 2025 and decline linearly to reach net zero by 2050."

(6) Line 203-204 mentions "foreign CDR purchase", what kind of mechanism is this and how is it set in the model?

(7) The paper mentions carbon price, what is the carbon price, and how is the carbon price mechanism of the model set?

(8) From the perspective of paper structure, there is a suggestion to merge Robustness check and Results benchmarking with discussion.

(9) There are still errors in the paper, such as line 488, below 1.5 should be 1.5 °C.

Reviewer #3 (Remarks to the Author):

The authors have addressed my comments adequately. I recommend acceptance.

Reviewer 1

Thank you for the detailed responses to my comments and for incorporating them into the revised manuscript. My concerns are sufficiently addressed in the revised manuscript. The added section explain mitigation deterrence and its role in this study carefully. Moreover, the influence of the SDR is put into perspective, both by your reply and the added text. The sensitivity analysis provides a significant contribution to the robustness of the results in regards to central parameters.

The authors thank the Reviewer for their careful attention to our paper. We appreciate the Reviewer's previous suggestions during the first round, which were very instrumental in strengthening our paper's contribution to the existing literature. We are also glad that our revisions were satisfactory to the Reviewer. Thank you.

Reviewer 2

According to the reviewer's comments, the author has made corresponding modifications to the paper. However, based on the current version, I do not think it is a high-level paper and I cannot accept its publication.

The authors thank the Reviewer for their careful attention to our paper and the constructive feedback. We appreciate the review highlighting further areas where we can strengthen our study design and writing. We have addressed each of the comments in detail, and we believe the revised manuscript would meet the quality standards of Nature Communications.

(1) Is the title "Deployment expectations of multi gigatonnes scale carbon dioxide removal could adaptively impact the global climate system" appropriate? It seems that the paper did not answer such a question. Firstly, the paper quantifies the impact of different scales of CDR technology on the energy-land-water-food system to limit warming to achieve net zero greenhouse gas emissions by 2050. Secondly, there are many indicators about the global climate system, including the

hydrosphere, atmosphere, etc. from a natural science perspective, but these related indicators are not reflected in the paper. If the author refers to the impact on low-carbon transformation, but the prerequisite for the paper is to achieve net zero greenhouse gas emissions by 2050, and the related result analysis only extends to 2050, and there is no impact on the period after 2050. Therefore, the relationship between the title and the research content is not very matching.

The authors discussed the Reviewer's concerns, and we agree that using the general term "climate system" overstated the scope of our research. Our work primarily focuses on impacts related to energy transitions, emission removal, as well as land and water use. As such, we have revised the paper title to ensure it aligns more closely with the content of our research:

“Deployment expectations of multi-gigatonnes scale carbon removal could have adverse impacts on Asia’s energy-water-land nexus”

(2) The abstract is too general to show the innovation of the paper, and the research conclusions are obvious, lacking new findings.

The authors acknowledge the reviewer's concerns, and we have revised the abstract to highlight the main **innovation** in this study:

Current integrated assessment models often treat carbon dioxide removal (CDR) and emissions reduction as equal mitigation options under an optimal carbon price approach. This oversimplified approach exaggerates CDR's importance in the near term by continually deferring urgent emission cuts under the assumption of future large-scale removals. Here, we for the first time, set explicit targets for carbon removal separate from decarbonization, and model different stylized CDR pathways for Asia. Our analysis quantifies the vastly different energy-land-water trade-offs faced under scenarios ranging from limited to extensive CDR reliance. We find that under a high CDR scenario, fossil fuel dependence persists, with residual CO₂ emissions reaching at least 8 GtCO₂/yr by 2050, compared to less than 1 GtCO₂/yr under moderate-to-low CDR reliance. Gross CO₂

removal surges to 12 GtCO₂/yr by 2050 in the high CDR case, compared to 0.5-2.3 GtCO₂ in the moderate to low CDR scenarios. Moreover, this high CDR pathway delays the achievement of net zero CO₂ emissions for several Asian countries, with some opting to purchase foreign offsets against rapid domestic emission reduction. The high CDR scenario leads to cropland reduction of 65-80% in major agricultural areas in the region. Our findings demonstrate that pursuing CDR under deep decarbonization has lower adverse impacts than under a less decarbonized system or without adequate CDR. This quantitative assessment provides critical insights to policymakers on balancing emission reductions and realistic negative emissions targets. Here, we show that while CDR is necessary, it is worth exploring strategies that reduce the need for excessive reliance on CDR while capitalizing on its advantages when it is most viable.

Our detailed response below also clarifies why our work provides new findings and key contributions that are not obvious concerning the quantification of CDR impacts on the energy-land-use system.

The scope of Nature Communications states that it is a multidisciplinary journal dedicated to publishing high-quality research in all areas of science. Papers published by the journal aim to represent important advances of significance to specialists within each field. We believe our research makes important advances that would be of significant interest to the broader climate change research community. Our key contributions are summarized below.

1. While previous studies on CDR deployment trade-offs have been primarily qualitative in nature, our study provides an important advancement in quantifying the energy-land-water trade-offs associated with varying levels of CDR reliance.

2. Based on their optimal carbon price pathways, CDR deployment in IAMs is often treated as an equal mitigation option with decarbonization, and this approach undermines urgent emission cuts¹⁻

³. There are urgent calls to separate CDR and emission reduction targets^{4,5}, and in this study, we

explicitly separate our CDR and decarbonization pathways in a manner different from existing studies^{3,6-13}. This approach, which is an important advancement in the field, would allow CDR to scale without undermining emission cuts.

3. Despite the need for separate emission reduction and removal targets, the net zero targets and Long-Term Strategies (LTS) of many countries and regions, including the US, China, and the EU, are yet to include such separation (as seen below for the US). Our LOW and MODERATE CDR scenarios are based on a separate CDR and decarbonization target. The results we have obtained, as compared to those in the PROMOTE scenario, show the importance of this separation and would be vital in the future LTS submissions of countries.

Source: Climate Action Tracker, 2024

4. Importantly, we have focused on the Asian region, where a significant share of future global CDR (up to 50%) would be concentrated^{6,14}. No previous study has comprehensively analyzed the energy-land-water trade-offs that different countries/regions in Asia are exposed to under

varying/uncertain future carbon removal. Asia's science-supported target and their achievement would be of high interest. We provide a quantitative basis for that.

5. Existing studies continue to predominantly model land-based CDR approaches globally, and countries with land limitations are at risk of such CDR approaches¹⁴. Many countries in Asia have limited land areas and biomass potential, as such they would need different CDR approaches. As shown in Figure 4 of the revised manuscript, we demonstrate how our six different CDR types (land-based, chemical, and geochemical) are distributed across Asia, and which ones would be more suitable for policymakers to pursue in their respective countries.

6. Stranded assets in the electric power sector are a major issue that carbon-locked-in countries, especially many in Asia, may face under climate goals. Yet, while such studies exist on the relationship between decarbonization and stranded assets¹⁵⁻¹⁷, no previous study has investigated the trade-offs in capital stock turnover in the electric power sector under varying future CDR. We add to these studies the impact of CDR and stranded assets, and we have revealed the synergies and trade-offs that exist.

(3) The logic of the paper is not clear, especially the stranded asset analysis in the power sector in the revised manuscript, which confuses me. In fact, if the author wants to analyze the impact of negative carbon technology on sustainable development from the perspective of negative carbon technology, it should establish a complete sustainable development framework, and discuss the social, economic, environmental, climate, and other impacts of negative carbon technology within this unified framework. However, the problem studied in the paper is not from the perspective of sustainable development, so what is the significance of adding this part of research? In addition, there is no method description for stranded asset in the Methods section. Stranded asset depends on the detailed information of thermal power plants. It is unknown how the author obtained parameters such as operating time of different power plants in different countries.

The authors acknowledge the Reviewer's concerns. Yes, we don't address all SDGs. We only address the subset of energy, water and land, and now we clarify that in the title and the abstract. Below is a general logical framework for our study to provide clarity.

Also, stranded assets are an important aspect of energy transitions, so we include that, and the reviewer is right that stranded asset descriptions are missing, and now we include them. Thanks for pointing out this omission. Our rationale for the inclusion of stranded assets and methodology description are summarized below:

Stranded assets in the electric power sector are a major issue that carbon-locked-in countries, especially many in Asia, may face under climate goals. Yet, while such studies exist on the

relationship between decarbonization and stranded assets¹⁵⁻¹⁷, no previous study has investigated the trade-offs in stranded assets in the electric power sector under varying and uncertain CDR. We add to these studies the impact of CDR and stranded assets, and we have revealed the synergies and trade-offs that exist.

Approach for estimating stranded assets

To quantify the annual capacity additions, stranded assets, and associated capital investments in the electricity sector, we employ the detailed methodology and data assumptions following the works of Iyer et al.¹⁶ and Binsted et al.¹⁵ (All data assumptions for the electricity sector in estimating the stranded assets are available in the supplementary material). GCAM tracks power plant capital stock, technology type, and vintage across the entire lifetime of each technology. The retirement of power plants can occur through two distinct pathways: 1) Natural retirement upon reaching the end of their designed physical lifetime, or 2) Profit-induced premature retirement when the continued operation of plants becomes less profitable, resulting in stranded assets (as per Equation 1).

$$G_{T,V,r}(t) = G_{T,V,r}(t - 1) \times [1 - f_{T,V,r}^N(t)] \times [1 - f_{T,V,r}^P(t)] \quad (1)$$

Where $G_{T,V,r}(t)$ represents the electricity generation by technology T and vintage V in region r in modelling period t . $f_{T,V,r}^N(t)$ and $f_{T,V,r}^P(t)$ are the fraction of natural and profit-induced retirement in modeling period t . The calculations for capacity and capital stock turnover in the electric power sector were done using Plutus package in R¹⁸.

An example of the natural retirement function is shown below.

Also, below is an example of the profit-induced retirement function.

(4) Line82-83, “Here, we show that deploying tens of gigatonnes of negative emissions by the mid-century could seriously impede climate goals and threatens sustainability across Asia” How can this sentence be derived and how to imbed climate goals? Moreover, sustainable development is a comprehensive concept. Can the analysis of only a few aspects in the paper support this statement?

We agree with the reviewer that this paper does not cover all aspects of sustainable development. As such, we have clarified throughout the paper that we are only addressing a subset

of sustainability related to clean energy, air pollution, land, and water use. In our revised manuscript, we explicitly refer to these specific impacts where needed.

(5) Suggest the author to clarify the reasons for the increase in NH₃ emissions in the low and medium scenarios of Line 181-182, as well as the relatively high scenarios?

We appreciate the Reviewer's suggestion, and we have clarified this part of our results in the revised manuscript as follows.

During this period, while most pollutants decrease across scenarios, NH₃ emissions show growth under the MODERATE and LOW scenarios. This increase in NH₃ emissions could be attributed to the fact that while the phasing out of fossil fuels, driven by the emission reduction target, leads to a reduction in other criteria air pollutants, factors contributing to NH₃ emissions remain relatively unaffected by the greenhouse gas constraint. These factors include persistent fertilizer use for crop and bioenergy crop cultivation, ongoing meat and dairy production, as well as other urban processes such as waste decomposition and emissions from cooking activities.

(6) The impact of emission reduction paths on emission reduction technologies is significant, and the Methodology section of the paper only considers one path, namely "total GHG emissions in the region peak before 2025 and decline linearly to reach net zero by 2050."

We acknowledge the Reviewer's concerns, and we explain below why we use a GHG emission reduction pathway. The carbon budget, which represents the cumulative anthropogenic CO₂ emissions¹⁹, is a commonly used approach in studies because it addresses comprehensively all sources of emissions through sectors, technologies, and fuels. This is in contrast to other emission reduction paths used in some studies, where the modeled policy focuses on fuel switching in selected sectors, energy-efficiency programs, renewable energy deployment, or biofuel standards. Such targeted modeling policies only track and reduce emissions in the specific sectors where they are implemented. For example, modeling biofuel standards as the sole emission reduction path

might address emissions in the transport sector, but not the electricity or building sectors. A policy focused only on renewable energy deployment may reduce power sector emissions but could shift energy demand and emissions to other sectors like transportation or industry if the broader energy system implications are not considered. In contrast, a net zero GHG policy allows for deep emission reduction across the entire economy. By using the net GHG emissions approach, our study can more comprehensively capture the impacts on the energy-land-water nexus across the entire economy, compared to approaches that focus on emission reductions through specific policies.

Furthermore, the window for achieving the Paris Agreement's aspirational goal of limiting global temperature change to 1.5°C above pre-industrial levels is rapidly closing. At the current rate of emissions, the 1.5°C carbon budget will be exhausted in the late 2020s¹⁹. Considering that the remaining carbon budget for 1.5°C could be completely exhausted before 2030 without temporary overshoot, the GHG constraint (carbon budget), where total emissions peak and decline linearly to a net zero point by mid-century is the most effective methodology to use to quickly produce deep emission cuts consistent with limiting warming to 1.5°C.

(7) Line 203-204 mentions "foreign CDR purchase", what kind of mechanism is this and how is it set in the model?

We appreciate the Reviewer's query and we have also clarified this mechanism in the main text for the benefit of prospective readers. To clarify, foreign CDR purchase is not something we set in the model, rather the model finds the most efficient allocation of emission reductions across countries, given their technological specifications. When a country falls short of net-zero target, we assume that the shortfalls need to be met by foreign CDR purchase.

In this context, "foreign CDR purchase" refers to countries' plans to buy CDR credits from other nations instead of achieving net-zero emissions domestically. Bhutan, Suriname, and Panama are among the few countries already at net-negative emissions due to their high forest cover²⁰. This

group of 'net-negative' countries has plans to sell forest carbon offset credits to other more polluting nations under the Paris Agreement ^{20,21}. For instance, in order to align with the Paris Agreement goals, Korea's Long-Term Strategy (LTS) includes provisions that allow the country to purchase carbon offsets from overseas ²². By purchasing these credits, the higher emitting countries can meet their net-zero target without international financial transfer ²⁰.

(8) The paper mentions carbon price, what is the carbon price, and how is the carbon price mechanism of the model set?

Thanks for raising this question. Carbon price is a jargon used by climate economists, and we have revised the manuscript to be more specific and clear about what we estimate here.

The marginal abatement cost, also known as the GHG price, refers to the cost incurred in reducing the last unit (e.g., one ton) of GHG emissions ²³. It represents the additional cost that would be required to achieve an incremental reduction in emissions.

The GHG price mechanism in the model is also briefly explained below and also made available in the Supplementary Information:

The model reduces greenhouse gases by placing a price on GHG emissions. This price then filters down through all the systems in the model and alters production and demand. For example, a price on carbon would put a cost on emitting fossil fuels. This cost would then influence the cost of producing electricity from fossil-fired power plants that emit CO₂, which would then influence their relative cost compared to other electricity-generating technologies and increase the price of electricity. The increased price of electricity would then make its way to consumers who use electricity, potentially decreasing its competitiveness relative to other fuels.

When a policy is initiated, such as a target to achieve net-zero emissions in 2050, the carbon price begins to increase and follows the steps described above to achieve the emission target across every modeling period.

(9) From the perspective of paper structure, there is a suggestion to merge Robustness check and Results benchmarking with discussion.

We appreciate the Reviewer's concerns but we believe merging these sections with the discussion would make the discussion section unnecessarily longer than it has to be. In the papers published in Nature Communications, Nature Climate Change, and other similar journals, the discussion section is typically a stand-alone section that re-iterates the key insights from the study. In our paper, the robustness check and results benchmarking sections are quite detailed and distinct from the typical discussion content. Merging these sections would imply that the "Discussion" heading does not accurately represent the contents that follow.

We have however implemented the Reviewer's suggestion for paper structure. We have sent the robustness check (validation) and sensitivity analysis results to the Method section instead, following similar manuscript organizations in recent publications in the journal.

(10) There are still errors in the paper, such as line 488, below 1.5 should be 1.5 °C..

We appreciate the Reviewer catching this error. We have made the correction, and have thoroughly double-checked the entire text to avoid such errors.

We are grateful to the Reviewer for the time taken to review and carefully comment on our manuscript. Suggestions and recommendations have significantly improved the paper.

Reviewer 3

The authors have addressed my comments adequately. I recommend acceptance.

The authors thank the Reviewer for their careful attention to our paper. We appreciate the Reviewer's previous suggestions during the first round, which were very instrumental in strengthening our paper's contribution to the existing literature. We are also glad that our revisions were satisfactory to the Reviewer. Thank you.

References

- (1) Lenzi, D. The Ethics of Negative Emissions. *Glob. Sustain.* **2018**, *1*, e7. <https://doi.org/10.1017/sus.2018.5>.
- (2) Rogelj, J.; Huppmann, D.; Krey, V.; Riahi, K.; Clarke, L.; Gidden, M.; Nicholls, Z.; Meinshausen, M. A New Scenario Logic for the Paris Agreement Long-Term Temperature Goal. *Nature* **2019**, *573* (7774), 357–363. <https://doi.org/10.1038/s41586-019-1541-4>.
- (3) Realmonte, G.; Drouet, L.; Gambhir, A.; Glynn, J.; Hawkes, A.; Köberle, A. C.; Tavoni, M. An Inter-Model Assessment of the Role of Direct Air Capture in Deep Mitigation Pathways. *Nat. Commun.* **2019**, *10* (1), 3277. <https://doi.org/10.1038/s41467-019-10842-5>.
- (4) McLaren, D. P.; Tyfield, D. P.; Willis, R.; Szerszynski, B.; Markusson, N. O. Beyond “Net-Zero”: A Case for Separate Targets for Emissions Reduction and Negative Emissions. *Front. Clim.* **2019**, *1*.
- (5) Höglund, R.; Mitchell-Larson, E.; Delerce, S. *How to Scale Carbon Removal without Undermining Emission Cuts*; Carbon Gap, 2023. <https://carbongap.org/how-to-scale-carbon-removal-without-undermining-emission-cuts/> (accessed 2023-10-04).
- (6) Fuhrman, J.; Bergero, C.; Weber, M.; Monteith, S.; Wang, F. M.; Clarens, A. F.; Doney, S. C.; Shobe, W.; McJeon, H. Diverse Carbon Dioxide Removal Approaches Could Reduce Impacts on the Energy–Water–Land System. *Nat. Clim. Change* **2023**. <https://doi.org/10.1038/s41558-023-01604-9>.
- (7) Calvin, K.; Wise, M.; Kyle, P.; Patel, P.; Clarke, L.; Edmonds, J. Trade-Offs of Different Land and Bioenergy Policies on the Path to Achieving Climate Targets. *Clim. Change* **2014**, *123* (3), 691–704. <https://doi.org/10.1007/s10584-013-0897-y>.
- (8) Grant, N.; Gambhir, A.; Mittal, S.; Greig, C.; Köberle, A. C. Enhancing the Realism of Decarbonisation Scenarios with Practicable Regional Constraints on CO₂ Storage Capacity. *Int. J. Greenh. Gas Control* **2022**, *120*, 103766. <https://doi.org/10.1016/j.ijggc.2022.103766>.
- (9) Kim, H.; McJeon, H.; Jung, D.; Lee, H.; Bergero, C.; Eom, J. Integrated Assessment Modeling of Korea’s 2050 Carbon Neutrality Technology Pathways. *Energy Clim. Change* **2022**, *3*, 100075. <https://doi.org/10.1016/j.egycc.2022.100075>.
- (10) Bistline, J. E. T.; Blanford, G. J. Impact of Carbon Dioxide Removal Technologies on Deep Decarbonization of the Electric Power Sector. *Nat. Commun.* **2021**, *12* (1), 3732. <https://doi.org/10.1038/s41467-021-23554-6>.
- (11) Fauvel, C.; Fuhrman, J.; Ou, Y.; Shobe, W.; Doney, S.; McJeon, H.; Clarens, A. Regional Implications of Carbon Dioxide Removal in Meeting Net Zero Targets for the United States. *Environ. Res. Lett.* **2023**, *18* (9), 094019. <https://doi.org/10.1088/1748-9326/aced18>.
- (12) Galán-Martín, Á.; Vázquez, D.; Cobo, S.; Mac Dowell, N.; Caballero, J. A.; Guillén-Gosálbez, G. Delaying Carbon Dioxide Removal in the European Union Puts Climate Targets at Risk. *Nat. Commun.* **2021**, *12* (1), 6490. <https://doi.org/10.1038/s41467-021-26680-3>.
- (13) Fuhrman, J.; Clarens, A.; Calvin, K.; Doney, S. C.; Edmonds, J. A.; O’Rourke, P.; Patel, P.; Pradhan, S.; Shobe, W.; McJeon, H. The Role of Direct Air Capture and Negative Emissions Technologies in the Shared Socioeconomic Pathways towards +1.5 °C and +2 °C Futures. *Environ. Res. Lett.* **2021**, *16* (11), 114012. <https://doi.org/10.1088/1748-9326/ac2db0>.
- (14) Yang, P.; Mi, Z.; Wei, Y.-M.; Hanssen, S. V.; Liu, L.-C.; Coffman, D.; Sun, X.; Liao, H.; Yao, Y.-F.; Kang, J.-N.; Wang, P.-T.; Davis, S. J. The Global Mismatch between Equitable Carbon Dioxide Removal Liability and Capacity. *Natl. Sci. Rev.* **2023**, *10* (12), nwad254. <https://doi.org/10.1093/nsr/nwad254>.

- (15) Binsted, M.; Iyer, G.; Edmonds, J.; Vogt-Schilb, A.; Arguello, R.; Cadena, A.; Delgado, R.; Feijoo, F.; Lucena, A. F. P.; McJeon, H.; Miralles-Wilhelm, F.; Sharma, A. Stranded Asset Implications of the Paris Agreement in Latin America and the Caribbean. *Environ. Res. Lett.* **2020**, *15* (4), 044026. <https://doi.org/10.1088/1748-9326/ab506d>.
- (16) Iyer, G.; Ledna, C.; Clarke, L.; Edmonds, J.; McJeon, H.; Kyle, P.; Williams, J. H. Measuring Progress from Nationally Determined Contributions to Mid-Century Strategies. *Nat. Clim. Change* **2017**, *7* (12), 871–874. <https://doi.org/10.1038/s41558-017-0005-9>.
- (17) Ou, Y.; Iyer, G.; McJeon, H.; Cui, R.; Zhao, A.; O’Keefe, K. T. V.; Zhao, M.; Qiu, Y.; Loughlin, D. H. State-by-State Energy-Water-Land-Health Impacts of the US Net-Zero Emissions Goal. *Energy Clim. Change* **2023**, *4*, 100117. <https://doi.org/10.1016/j.egycc.2023.100117>.
- (18) Zhao, M.; Binsted, M.; Wild, T.; Khan, Z.; Yarlagadda, B.; Iyer, G.; Vernon, C.; Patel, P.; Da Silva, S.; Calvin, K. Plutus: An R Package to Calculate Electricity Investments and Stranded Assets from the Global Change Analysis Model (GCAM). *J. Open Source Softw.* **2021**, *6* (65), 3212. <https://doi.org/10.21105/joss.03212>.
- (19) Lamboll, R. D.; Nicholls, Z. R. J.; Smith, C. J.; Kikstra, J. S.; Byers, E.; Rogelj, J. Assessing the Size and Uncertainty of Remaining Carbon Budgets. *Nat. Clim. Change* **2023**. <https://doi.org/10.1038/s41558-023-01848-5>.
- (20) Dunne, D. *Explainer: Why some countries are aiming for “net-negative” emissions*. Carbon Brief. <https://www.carbonbrief.org/explainer-why-some-countries-are-aiming-for-net-negative-emissions/> (accessed 2024-04-10).
- (21) Spring, J.; Spring, J. Exclusive: Suriname Aims to Be First to Sell Paris Agreement Carbon Credits, Adviser Says. *Reuters*. September 14, 2023. <https://www.reuters.com/sustainability/sustainable-finance-reporting/suriname-aims-be-first-sell-paris-agreement-carbon-credits-adviser-2023-09-13/> (accessed 2024-04-12).
- (22) The Government of Republic of Korea. *2050 Carbon Neutral Strategy of the Republic of Korea towards a Sustainable and Green Society*; 2020. https://unfccc.int/sites/default/files/resource/LTS1_RKorea.pdf.
- (23) IPCC. *Climate Change 2022: Mitigation of Climate Change. Contribution of Working Group III to the Sixth Assessment Report of the Intergovernmental Panel on Climate Change*; [P.R. Shukla, J. Skea, R. Slade, A. Al Khourdajie, R. van Diemen, D. McCollum, M. Pathak, S. Some, P. Vyas, R. Fradera, M. Belkacemi, A. Hasija, G. Lisboa, S. Luz, J. Malley, (eds.)]. Cambridge University Press, Cambridge, UK and New York, NY, USA, 2022.

REVIEWERS' COMMENTS

Reviewer #2 (Remarks to the Author):

It can be accepted.

Reviewer 2

It can be accepted.

The authors thank the Reviewer for their careful attention to our paper and the constructive feedback. We appreciate the Reviewer's suggestions throughout the entire process which have helped improved our manuscript. We are glad that our last revisions met the expectations of the Reviewer.